# Continuous and autonomous snow water equivalent measurements by a cosmic ray sensor on an Alpine glacier

Rebecca Gugerli[1], Nadine Salzmann[1], Matthias Huss[1,2], and Darin Desilets[3]

[1]Department of Geosciences, University of Fribourg, Fribourg, Switzerland
[2]Laboratory of Hydraulics, Hydrology and Glaciology (VAW), ETH Zurich, Zurich, Switzerland
[3]Hydroinnova LLC, Albuquerque, NM, USA

**Correspondence:** Rebecca Gugerli (rebecca.gugerli@unifr.ch)

**Abstract.** Snow water equivalent (SWE) measurements of seasonal snowpack are crucial in many research fields. Yet accurate measurements at a high temporal resolution are difficult to obtain in high mountain regions. With a cosmic ray sensor (CRS), SWE can be inferred from neutron counts. We present the analyses of temporally continuous SWE measurements by a CRS on an Alpine glacier in Switzerland (Glacier de la Plaine Morte) over two winter seasons (2016/17 and 2017/18), which differed markedly in the amount and timing of snow accumulation. By combining SWE with snow depth measurements, we calculate the daily mean density of the snowpack. Compared to manual field observations from snow pits the autonomous measurements overestimate SWE by +2%±13%. Snow depth and the bulk snow density deviate from the manual measurements by ±6% and ±9% respectively. The CRS measured with high reliability over two winter seasons and is thus considered a promising method to observe SWE at remote alpine sites. We use the daily observations to classify winter season days into those dominated by accumulation (solid precipitation, snow drift), ablation (snow drift, melt) or snow densification. For each of these process-dominated days the prevailing meteorological conditions are distinct. The continuous SWE measurements were also used to define a scaling factor for precipitation amounts from nearby meteorological stations. With this analysis, we show that a best-possible constant scaling factor results in cumulative precipitation amounts that differ by a mean absolute error of less than 80 mm w.e. from snow accumulation at this site.

## 1 Introduction

The evolution and amount of seasonal snow accumulation in high mountain regions is a key parameter in many climate-related research fields such as glaciology or hydrology and climate change impacts, risks and adaptation. Changes in snow accumulation in mountain areas caused by climate change are expected to have major impacts on water supply for adjacent lowlands (Barnett et al., 2005; Viviroli et al., 2007, 2011), hydropower production (Ali et al., 2018) or winter tourism (Marty et al., 2014; Sturm et al., 2017). In addition, information of the amount of water stored within the annual snowpack (snow water equivalent, SWE) in high mountain regions is crucial for avalanche warning (Castebrunet et al., 2014), flood prevention (Jörg-Hess et al., 2015), or mass balance calculations of glaciers (Sold et al., 2013; Pulwicki et al., 2018). Despite the high demand for accurate SWE measurements in high mountain regions, reliable and temporally continuous measurements of SWE are still difficult to obtain. In particular, the cold and windy conditions pose the main challenge for accurate measurements

(Sevruk et al., 2009; Rasmussen et al., 2012; Kinar and Pomeroy, 2015). The complex topography and limited accessibility add further challenges.

In this study we focus on providing temporally continuous and autonomous observations of SWE in glacierized high mountain regions to improve our understanding of the seasonal evolution of the snowpack.

## 1.1 State-of-the-art snow accumulation observations

A wide range of different devices are used to measure snow accumulation (Kinar and Pomeroy, 2015; Pirazzini et al., 2018), each with its advantages and clear tradeoffs. Manual in situ field measurements with snow pits and snow probes can provide reliable data but have a low temporal resolution. Such measurements are also invasive, laborious, and logistically complicated for remote sites.

According to Pirazzini et al. (2018) snow gauges, a rain-gauge adapted for solid precipitation, are often used in Europe. However, they are known to carry large uncertainties in the extreme environments of high mountains through undercatch and post-event thawing (e.g. Goodison et al., 1998; Rasmussen et al., 2012; Martinaitis et al., 2015; Pollock et al., 2018). Current instruments relying on the mass or pressure of overlying snow (e.g. snow pillows and snow scales) are not well suited for high mountain regions because they require a large flat surface (e.g. Egli et al., 2009; Kinar and Pomeroy, 2015). In addition, ice bridging produce large errors (e.g. Sorteberg et al., 2001; Johnson and Schaefer, 2002).

Other in situ devices include ground-penetrating radar (GPR) and sub-snow GPSs. Upward-looking GPR systems are installed below the snowpack and provide information about the snow stratigraphy (Heilig et al., 2009) and snow depth (SD, Heilig et al., 2010; Schmid et al., 2014). Combined with a low-cost GPS, Schmid et al. (2015) derived the liquid water content, SD and SWE independently from additional information and mast poles, making the system suitable for avalanche-prone slopes. Recent studies present sub-snow low-cost GPS as a promising method to continuously derive SWE (Steiner et al., 2018; Henkel et al., 2018; Steiner et al., 2019; Koch et al., 2019). This method uses two GPS antennas one of which is placed below and the other above the snowpack. Because the GPS signals are influenced when traveling through the snowpack, the difference in received signals can be used to quantify SWE, SD and liquid water content. GPS signals are freely available but the signal strength may be limited in high mountain regions depending on slope aspect and location (Koch et al., 2019).

Spaceborne sensors can provide observations of snow cover, SWE and SD with a large spatial coverage. However, these observations often have a low spatial resolution and estimates of SWE are affected by snow properties such as microstructure and liquid water content (Clifford, 2010; Dietz et al., 2012). In addition, uncertainties are increased for complex topographies (Smith and Bookhagen, 2016) and deep snowpacks (Smith and Bookhagen, 2018).

Other approaches use empirically derived or physically calculated bulk snow densities with additional meteorological parameters to calculate SWE from continuous SD measurements. Empirical models often estimate a bulk snow density, which allows the calculation of SWE if combined with SD data (e.g. Jonas et al., 2009; Sturm et al., 2010). More recently, Hill et al. (2019) proposed an empirical model to derive SWE from SD measurements in regions where no automatic weather station (AWS) data is available. For avalanche forecasting operational models are usually physically-based (e.g. Crocus, Vionnet et al., 2012, SNOWPACK, Lehning et al., 1999) and require high-quality meteorological data to derive accurate snow properties. Such

model-based approaches are sensitive to errors in the input data. For example, erroneous precipitation observations (Raleigh et al., 2015) and/or uncertainties in modeled snow density may influence the results significantly (Raleigh and Small, 2017).

A simplified approach utilizes precipitation observations from nearby AWS and accounts for the bias in cumulative seasonal precipitation through use of a temporally constant scaling factor derived from on-glacier point SWE measurements at the end of winter. This approach is used for the operational evaluation of the winter mass balance on Swiss glaciers (GLAMOS, 2018) where seasonal manual measurement of SWE are combined with readily available precipitation data. Despite the heterogeneity of precipitation and the influence of preferential deposition and snow drift, the simplified approach has provided reasonable results for the purpose of glacier mass balance observations (see e.g. Huss et al., 2009, 2015; Sold et al., 2016).

## 1.2 Cosmic ray sensor

The cosmic ray sensor (CRS) is a device to measure snow accumulation temporally continuous. The method relies on the attenuation of natural radiation by snow. A CRS counts the number of fast neutrons and the one used in this study is installed at ground level, and is allowed to get buried by snow. On the surface of a glacier, most of the fast neutrons originate from the atmosphere and are moderated by the hydrogen atoms contained in water (whether in solid or liquid form). Hence, the neutron counting rate is negatively correlated to the number of hydrogen atoms above the sensor. A CRS was deployed by Kodama et al. (1975) and Kodama (1980) in the 1970s and showed promising results with an error less than 7% for cosmic-ray-derived SWE measurements compared to manual measurements. Almost 20 years later, the Électricité de France developed their own CRS and integrated these in a mountain monitoring network in order to manage hydroelectric power plants (Paquet and Laval, 2005; Paquet et al., 2008). In 2013, this monitoring network counted 37 sites in the French Alps and the Pyrenees (Gottardi et al., 2013).

Recent studies have investigated the potential of SWE measurements with CRS installed above the snowpack. These provide a larger footprint with a radius on the order of tens to hundreds of meters (Sigouin and Si, 2016; Schattan et al., 2017). Independently of the sensor's deployment above or below the snowpack, the SWE measurements are known to be influenced by changes in soil moisture through snow melt (Kodama, 1980; Paquet and Laval, 2005; Sigouin and Si, 2016). A shield was thus added to the invasive CRS to prevent influences from increases in soil moisture from the surrounding ground (Paquet and Laval, 2005). Schattan et al. (2017) state that with the non-invasive sensor the effect is negligible for deep snowpacks. These influences are avoided by placing the CRS on an ice surface such as a polar ice sheet or a mountain glacier. In the recent study by Howat et al. (2018), the CRS was deployed below the snowpack on the Greenland Ice Sheet. With almost 24 months of measurements, they find an instrument precision of approximately 0.7% and a good agreement with manual measurements.

## 1.3 Study objectives

In this study, we investigate the applicability of a CRS installed below the snowpack to derive continuous SWE observations on an Alpine glacier in Switzerland (Glacier de la Plaine Morte). More specifically, we (i) analyse the CRS performance by comparing its SWE estimates to manual field observations. With the continuous observations of SWE and SD we (ii) analyze the evolution of snow density over the course of a winter season including the influence of meteorological conditions. Finally,

we use the continuous observations to (iii) assess the performance of scaling readily available precipitation observations of nearby AWS and gridded precipitation data with a temporally constant factor.

## 2 Study site and data

### 2.1 Study site

5 Our study site is located on the Glacier de la Plaine Morte (in the following: Plaine Morte) in Switzerland, where we deployed a subsurface CRS along with an AWS at an elevation of 2690 m a.s.l. (Fig. 1). Plaine Morte is situated on the divide between the Bernese Alps in the North and the Rhône valley in the South (Huss et al., 2013) and is surrounded by mountain peaks with elevations from 2926 m.a.s.l (Pointe de la Plaine Morte) up to 3244 m a.s.l. (Wildstrubel, see Fig. 1).

With a surface area of 7.4 $km^2$ and a particularly low elevation gradient, Plaine Morte is the largest plateau glacier in 10 the European Alps. Most of its surface is located between 2650 m a.s.l. and 2800 m a.s.l. (GLAMOS, 2018). Due to lack of elevation gradient, the equilibrium line can be located either above or below the glacier surface, rendering it either completely snow-free or snow-covered at the end of summer. For the same reason, the winter snow distribution shows only a small spatial variability (GLAMOS, 2018) and the surface velocity is low (2-5 m per year according to Huss et al., 2013).

Since its inclusion in the glacier monitoring network of Switzerland, the annual glacier-wide mass balance for the hydrolog-15 ical years between 2009 and 2019 has been negative with an average loss of 1477 mm w.e. Average glacier-wide winter mass balance was 1338 mm w.e. between 2010 and 2019 (GLAMOS, 1881-2018).

In October 2016, we installed an AWS on Plaine Morte (46° 22.8' N, 7°29.7' E, 2690 m a.s.l., Fig. 1) with sensors to measure SD, air temperature, humidity, air pressure and shortwave radiation (the latter added in October 2017). The CRS (SnowFox[TM] provided by Hydroinnova LLC, Albuquerque, NM, USA) is also connected to the station. We conducted 11 field campaigns 20 over two winter seasons to measure SD and SWE manually. Additionally, we use observational and gridded meteorological data provided by the Federal Office of Meteorology and Climatology (MeteoSwiss) for comparison and for best-possible data completion, as described below.

### 2.2 Automatic weather stations

We installed a five meter tall mast on the bare ice of Plaine Morte on which we mounted all sensors (see Table 1) at 4.8 m height 25 above the glacier surface. These sensors measured continuously at an hourly interval during two winter seasons (20 October 2016 to 29 July 2018). The CRS lies on the bare ice, i.e. below the snowpack, at approximately 8 m horizontal distance from the mast to limit impacts caused by potential maintenance work.

Precipitation data for comparison to snow accumulation are taken from (i) the federal network of weather stations in Switzerland (SwissMetNet, Table 2) of which we selected those stations located close to Plaine Morte and with high data quality, and 30 (ii) a gridded precipitation product (RhiresD). We did not include precipitation data from the high-elevation weather station

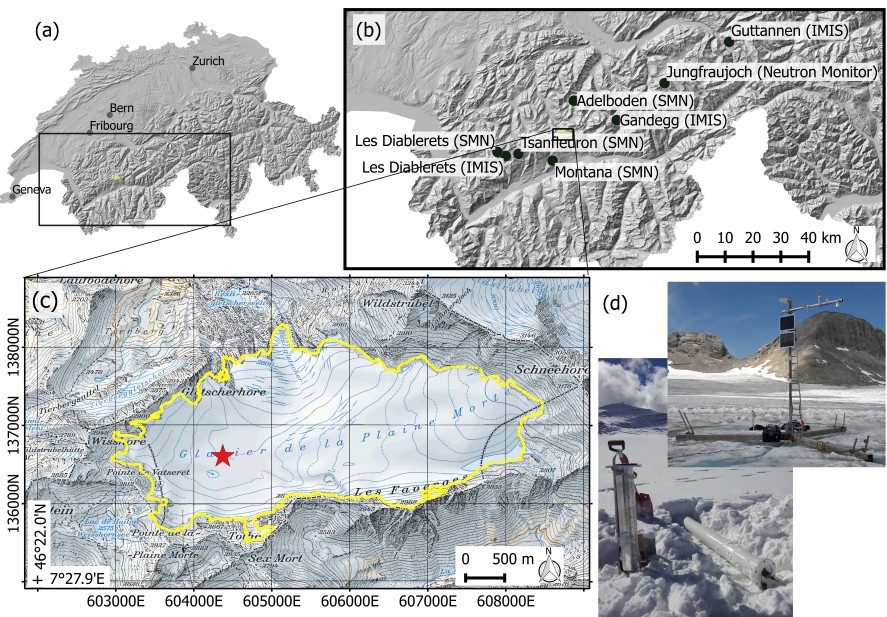

**Figure 1.** (a) Map of Switzerland. (b) Map of the excerpt marked in (a), and all weather stations used in this study (black dots). (c) Topographic map of Plaine Morte with the red star indicating the location of the AWS with the CRS (see (d), 46°22.8'N, 7°29.7'E). The yellow contour represents the current outline of Plaine Morte (Fischer et al., 2014). The coordinates correspond to the Swiss coordinate system (EPSG: 21781). (Maps provided by Swisstopo).

**Table 1.** Sensors installed at Plaine Morte.

| Name | Distributor | Parameter |
|---|---|---|
| CNR4 | Kipp & Zonen | shortwave radiation |
| CS215 | Campbell Scientific | air temperature, relative humidity |
| UMB Ventus | Lufft | air pressure, wind speed, wind direction |
| SnowFox<sup>TM</sup> (CRS) | Hydroinnova | snow water equivalent (fast neutrons) |
| SR50A | Campbell scientific | snow depth |

network in the Swiss Alps (IMIS, intercantonal measurement and information system, SLF Data, 2015) as the ones with a pluviometer have considerable data gaps during the winter season.

The gridded precipitation product, RhiresD, uses rain-gauge measurements from around 400 automatic as well as manual observations. These observations (not available in real time) are quality-checked prior to their processing. The observations are spatially analysed, pre-processed and interpolated to a 1×1 km grid at daily resolution covering the Swiss territory (MeteoSwiss, 2013). The main sources of uncertainty arise from the interpolation, the rain-gauge measurements, the grid spacing

**Table 2.** Table with three meteorological stations with precipitation measurements and three grid cells of the gridded precipitation product. For the gridded precipitation, the coordinate in EPSG 21781 represents the centers of the corresponding pixel (see Fig. 1).

| Station name | Coordinates (WGS 84) [EPSG 21781] | Elevation [m a.s.l.] | Source |
|---|---|---|---|
| Adelboden | 46°30'N, 7°34'E | 1322 | SwissMetNet |
| Montana | 46°18'N, 7°28'E | 1427 | SwissMetNet |
| Tsanfleuron | 46°19'N, 7°18'E | 2052 | SwissMetNet |
| grid cell 1 (1 km$^2$) | 46°22'N, 7°30'E [605500E, 136500N] | 2299 | RhiresD |
| grid cell 2 (1 km$^2$) | 46°22'N, 7°29'E [604500E, 136500N] | 2579 | RhiresD |
| grid cell 3 (1 km$^2$) | 46°23'N, 7°29'E [604500E, 137500N] | 2579 | RhiresD |

and its effective resolution, and the temporal variation of the number of stations. For further information, the reader is referred to the technical document provided by MeteoSwiss (MeteoSwiss, 2013). We extracted daily precipitation estimates of the three grid points closest to the position of the CRS (Table 2 and Fig. 1c).

## 2.3 Field data

Over the two winter seasons 2016/17 and 2017/18, we conducted 11 field campaigns to obtain comparative data. During two of the campaigns (20 October 2016 and 5 December 2017) we installed the CRS which disturbed the snowpack. Hence, the measurements of these two campaigns are used only to account for the already fallen snow on the glacier.

During the field campaigns we measured SWE using snow pits and snow tube sampling (e.g. Cogley et al., 2011; Kinar and Pomeroy, 2015) and SD by snow probing. The snow pits were dug within approximately 15 meters of the station, but each time at a different location to avoid sampling of a disturbed snowpack.

## 3 Methods

### 3.1 Filling measurement gaps

This study covers two sequential but distinct winter seasons (20 October 2016 to 29 July 2018). During summer 2017, the AWS on Plaine Morte measured only wind speed, wind direction, temperature and relative humidity.

In winter 2017/18, unusually high amounts of snow buried most of the mast causing several interruptions of measurements. The time spans of the data gaps differ for certain sensors because of their measurement characteristics. The SD sensor, for

example, requires a minimal distance of 0.5 m to the target surface (Campbell Scientific, 2016) and has the longest data gap. Another issue was the power consumption of the station when the solar panels became buried by snow, too. To conserve power, we deactivated the heated wind sensor (highest energy consumption), which measures wind speed, wind direction, and air pressure. Furthermore, we disregarded wind speed, temperature and relative humidity from 10 March 2018 to 17 April 2018

because of the proximity of the sensors to the snow surface.

The CRS, in contrast, measured continuously over the two winter seasons with the exception of a short period end of April 2018. After fixing a faulty connection, the CRS continued measuring without need for further maintenance.

To fill the data gaps we correlated our measurements of the Plaine Morte site with data from the IMIS network and a selection of stations from SwissMetNet. For SD, air pressure and wind speed, we chose the station with the highest correlation (Table 3).

As SD is an accumulated time series, we correlated the daily change in SD. We did not fill the gap of wind direction because all correlations were below 0.45 at hourly as well as daily resolution. The mean bias in Table 3 is used to adjust the reference data to the Plaine Morte station. The standard deviation of the mean bias represents the absolute uncertainty of the parameters during the interpolated time period.

### 3.2   Calculating SWE from neutron counts

The CRS records the total number of cosmic ray neutrons integrated over a set time period, in this case one hour. The neutron count rate expressed in counts per hour (cph) is then used to infer SWE.

We process the raw neutron count rate as follows: To eliminate spurious changes in the count rate, neutron counts are excluded if the hourly count differs more than 20% from an 6-hour moving average. As presented in previous literature (e.g. Zreda et al., 2012; Hawdon et al., 2014; Sigouin and Si, 2016; Andreasen et al., 2017), we correct the neutron count rate

($N_{\mathrm{raw},i}$) for time step $i$ for variations in solar activity ($F_{\mathrm{s},i}$) and more importantly for changes in situ air pressure ($F_{\mathrm{p},i}$) with

$$N_i = N_{\mathrm{raw},i} \cdot F_{\mathrm{s},i} \cdot F_{\mathrm{p},i} \tag{1}$$

Variations in solar activity are quantified with the aid of a reference station, which is not buried in the snow. As a reference station we use the neutron monitor at Jungfraujoch (JUNG, www.nmdb.eu, see Fig. 1) which is located only 40 km from our site. The correction factor $F_{\mathrm{s}}$ is determined as

$$F_{\mathrm{s},i} = \beta \cdot \left( \frac{F_{\mathrm{inc},i}}{F_{\mathrm{inc},0}} - 1 \right) + 1 \tag{2}$$

where variable $F_{\mathrm{inc},i}$ represents the incoming neutron flux at Jungfraujoch (JUNG) at time interval $i$ and $F_{\mathrm{inc},0}$ represents the incoming neutron flux at an arbitrary reference time period. The adjustment factor $\beta$ depends on the difference in geomagnetic latitude and site elevation between the glacier site and the reference site (Desilets et al., 2006; Hawdon et al., 2014; Andreasen et al., 2017). The manufacturer has provided a value of 0.95 for our site. The adjustment is negligibly small because the study

site is located geographically close to the neutron monitor at Jungfraujoch.

Air pressure is directly measured at the study site. The correction factor $F_{\mathrm{p},i}$ is obtained by

$$F_{\mathrm{p},i} = exp \left( \frac{p_i - p_0}{L} \right) \tag{3}$$

**Table 3.** Time periods of data gaps with reference periods for correlation and correlation coefficients. The mean bias shows the average difference (and its standard deviation) between the reference stations and AWS at Plaine Morte. All stations are shown in Fig. 1

| Parameter | Data gap | Reference station Name, Network Coordinates | Correlation periods | Correlation mean bias |
|---|---|---|---|---|
| snow depth | 20 Jan 2018 to 4 May 2018 | Gandegg/ Laucherenalp SLFGA2, IMIS 46°26'N, 7°46'E, 2717 m a.s.l. | 4 Nov 2016 to 13 Jul 2017 27 Oct 2017 to 19 Jan 2018 4 May 2018 to 26 Jul 2018 | 0.86 (daily) 0.1±6.0 cm |
| air pressure | 22 Jan 2018 to 10 Mar 2018 | Les Diablerets DIA, SwissMetNet 46°20'N, 7°12'E, 2964 m a.s.l. | 1 Nov 2016 to 22 Jan 2018 10 Mar 2018 to 1 Mar 2018 | 0.996 (hourly) -23.96±0.6 hPa |
| wind speed | 22 Jan 2018 to 17 Apr 2018 | Guttannen/ Homad SLFGU2, IMIS 46°41'N, 8°17'E, 2110 m a.s.l. | 1 Jan 2018 to 22 Jan 2018 17 Apr 2018 to 30 Sep 2018 | 0.87 (daily) 0.1±1.0 m s$^{-1}$ |
| temperature | 10 Mar 2018 to 17 Apr 2018 | Les Diablerets SLFDIA, IMIS 46°19'N, 7°15'E, 2575 m a.s.l. | 1 Jan 2018 to 10 Mar 2018 17 Apr 2018 to 30 Sep 2018 | 0.96 (hourly) 1.8±2.4 °C |
| relative humidity | 10 Mar 2018 to 17 Apr 2018 | Les Diablerets SLFDIA, IMIS 46°19'N, 7°15'E, 2575 m a.s.l. | 1 Jan 2018 to 10 Mar 2018 17 Apr 2018 to 30 Sep 2018 | 0.78 (daily) -7.8±8.5 % |

The mass attenuation length $L$ is assumed to be 132 hPa for our study site and depends on latitude and atmospheric depth (Desilets et al., 2006). The observed hourly pressure values are represented by $p_i$ while $p_0$ stands for a reference pressure.

For the reference period, we chose a 24-hour time frame between 12 June 2017 at 10 UTC and 13 June 2017 at 10 UTC. The reference variables ($N_0$, $F_{\text{inc},0}$, $p_0$) correspond to the median value during the reference period (Table 5).

5    To calculate SWE, we use the relative neutron count ($N_{\text{rel},i}$, Eq. 4), i.e. the neutron count ($N_i$) divided by a reference count ($N_0$).

$$N_{\text{rel},i} = \frac{N_i}{N_0} \tag{4}$$

The relative neutron count is then used to derive SWE with the non-linear equation

$$SWE_i = -\frac{1}{\Lambda} \cdot \ln N_{\text{rel},i} \tag{5}$$

**Table 4.** The constant parameters of Eq. 6. The fitted parameters $a_1$, $a_2$ and $a_3$ are without unit.

| Parameters | Fit |
|---|---|
| $\Lambda_{\text{max}}$ | 114.4 cm |
| $\Lambda_{\text{min}}$ | 14.1 cm |
| $a_1$ | 0.313 |
| $a_2$ | 0.082 |
| $a_3$ | 1.117 |

The variable $\Lambda_i$ is the effective attenuation length given by

$$\Lambda_i = \frac{1}{\Lambda_{\text{max}}} + \left(\frac{1}{\Lambda_{\text{min}}} - \frac{1}{\Lambda_{\text{max}}}\right) \cdot \left(1 + exp\left(-\frac{N_{\text{rel},i} - a_1}{a_2}\right)\right)^{-a_3} \tag{6}$$

The empirical parameters $\Lambda_{\text{min}}$, $\Lambda_{\text{max}}$, $a_1$, $a_2$ and $a_3$ (Table 4) were provided by the manufacturer for use on glaciers and were also used by Howat et al. (2018). Note that the parameters $\Lambda_{min}$ and $\Lambda_{max}$ are respectively the asymptotic values of the effective attenuation lengths for low and high SWE values, and that the parameters $a_1$, $a_2$ and $a_3$ define the curvature of a sigmoidal function.

   In this study, we report daily estimates of SWE. The direct observations, however, are based on hourly values. Therefore, we integrated the mean daily neutron counts of $N_{\text{raw},i}$, $N_0$, $F_{\text{inc},i}$, $F_{\text{inc},0}$ over 24 hours and took the mean daily pressure for $p_i$ to calculate SWE.

## 3.3 Calculating snow density and daily changes in SWE, SD and snow density

The bulk snow density ($\rho_{crs\_sr}$, in kg m$^{-3}$) is derived from daily SWE ($SWE_{crs}$, in mm w.e. or kg m$^{-2}$, Fierz et al., 2009) and daily SD measurements ($SD_{sr}$, in m) according to

$$\rho_{crs\_sr} = \frac{SWE_{crs}}{SD_{sr}} \tag{7}$$

   The temporal resolution of one day allows the determination of daily changes in SD, SWE and the bulk snow density. These daily changes are calculated as the difference between two consecutive days. We filtered out days where daily changes where lower than the uncertainty estimates.

## 3.4 Estimating the uncertainty of the CRS

The calculated SWE is determined by the corrected neutron count relative to when the CRS is uncovered by snow ($N_{\text{rel},i}$, Eq. 4). We base our error propagation on all corrections applied to the raw neutron count. We assemble Eq. 1-4 into

$$N_{\text{rel},i} = N_{\text{raw},i} \cdot \left(\beta \cdot \left(\frac{F_{\text{inc},i}}{F_{\text{inc},0}} - 1\right) + 1\right) \cdot exp\left(\frac{p_i - p_0}{L}\right) \cdot \frac{1}{N_0} \tag{8}$$

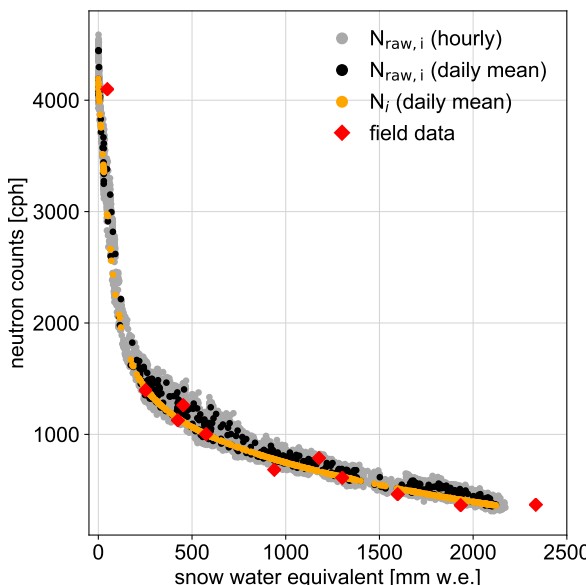

**Figure 2.** Relation between SWE and the neutron count rate. Grey dots represent the uncorrected hourly neutron counts and black dots the uncorrected daily means. The orange dots represent the corrected daily means. Red dots show SWE from the field data and the corresponding neutron counts of the field work days.

The raw neutron count ($N_{\mathrm{raw},i}$), the incoming neutron flux ($F_{\mathrm{inc},i}$) and air pressure ($p_i$) change with time, but remain independent from each other. Following the rules of error propagation of a non-linear equation, we approximate the uncertainty in $N_{\mathrm{rel},i}$ as

$$
5 \quad \sigma^2_{N_{\mathrm{rel},i}} \approx \left( \frac{\partial N_{\mathrm{rel},i}}{\partial N_{\mathrm{raw},i}} \right)^2 \cdot \sigma^2_{N_{\mathrm{raw},i}} + \left( \frac{\partial N_{\mathrm{rel},i}}{\partial N_0} \right)^2 \cdot \sigma^2_{N_0}
$$

$$
+ \left( \frac{\partial N_{\mathrm{rel},i}}{\partial F_{\mathrm{inc},i}} \right)^2 \cdot \sigma^2_{F_{\mathrm{inc},i}} + \left( \frac{\partial N_{\mathrm{rel},i}}{\partial F_{\mathrm{inc},0}} \right)^2 \cdot \sigma^2_{F_{\mathrm{inc},0}} + \left( \frac{\partial N_{\mathrm{rel},i}}{\partial \beta} \right)^2 \cdot \sigma^2_{\beta}
$$

$$
+ \left( \frac{\partial N_{\mathrm{rel},i}}{\partial p_i} \right)^2 \cdot \sigma^2_{p_i} + \left( \frac{\partial N_{\mathrm{rel},i}}{\partial p_0} \right)^2 \cdot \sigma^2_{p_0} + \left( \frac{\partial N_{\mathrm{rel},i}}{\partial L} \right)^2 \cdot \sigma^2_{L} \quad (9)
$$

The uncertainty $\sigma^2_{N_{\mathrm{rel},i}}$ is then propagated through Eq. 5 to estimate the uncertainty $\sigma_{\mathrm{crs},i}$

$$
\sigma_{\mathrm{crs},i} \approx \sqrt{ \left( \frac{\partial SWE_i}{\partial N_{rel,i}} \right)^2 \cdot \sigma^2_{N_{rel,i}} } \quad (10)
$$

10 Since the uncertainties are not always known, we assume rather generous estimates for the uncertainties of all correction factors. Table 5 provides an overview of uncertainty estimates for all components.

For all neutron count rates ($N_{\mathrm{raw},i}$, $N_0$, $F_{\mathrm{inc},0}$, $F_{\mathrm{inc},i}$), we assume Poisson statistics, which gives the uncertainty as the square root of the neutron counts (e.g. Zreda et al., 2012). With the integration over a time period $t$, the uncertainty is reduced by

**Table 5.** Compilation of all direct observations and constants as well as the associated uncertainties $\sigma$ at the hourly and daily scale. The units cph and cps stand for counts per hour and second, respectively. Brackets show the minimum and maximum within the time series.

| Variables | hourly values | $\sigma$ (hourly) | $\sigma$ (daily) |
|---|---|---|---|
| $N_{\mathrm{raw},i}$ | [354; 4450] cph | $\sqrt{N_{\mathrm{raw},i}}$ cph | $\sqrt{\frac{N_{\mathrm{raw},i}}{24}}$ cph |
| $N_0$ | 4143 cph | 64 cph | 13 cph |
| $F_{\mathrm{inc},i}$ | [184; 195] cps | $\sqrt{\frac{F_{\mathrm{inc},i}}{3600}}$ cps | $\sqrt{\frac{F_{\mathrm{inc},i}}{86400}}$ cps |
| $F_{\mathrm{inc},0}$ | 191 cps | 0.2 cps | 0.1 cps |
| $\beta$ | 0.95 | 0.03 | 0.03 |
| $p_i$ | [708; 747] hPa | 0.1 hPa | 0.1 hPa |
| $p_0$ | 739 hPa | 0.1 hPa | 0.1 hPa |
| $L$ | 132 hPa | 2 hPa | 2 hPa |

$t^{-0.5}$ (Schrön et al., 2018). While the relative uncertainty in $N_{\mathrm{raw},i}$ varies between 1.5%-5.3% for hourly observations, it varies between 0.3%-1% for the integrated daily estimates of our study.

The incoming radiation measured at Jungfraujoch has a low statistical uncertainty as its precision is high with around 190 counts per second. However, incoming radiation is corrected by an adjustment factor ($\beta$, Eq. 2) which is rather small for our
site. Therefore, we assume also a small uncertainty of 0.03 for $\sigma_\beta$.

The uncertainty in air pressure ($\sigma_{p_i}$, $\sigma_{p_0}$) is based on the instrumental precision of 0.1 hPa (Lufft, 2019). For the mass attenuation length $L$, we use 132 hPa. An applied uncertainty of of $\pm 2$ hPa corresponds to the difference of shielding depths from latitudes north and south of Switzerland as shown in Fig.1 of Andreasen et al. (2017).

To render the error propagation more robust, we calculated $\sigma_{\mathrm{crs},i}$ using two different time resolutions. We additionally
created a synthetic data set for both time resolutions. For the synthetic data set, we varied the time-dependent variables ($N_{\mathrm{raw},i}$, $p_i$, $F_{\mathrm{inc},i}$) uniformly within their observed minima and maxima values. At the hourly resolution it encompasses $4.8 \cdot 10^5$ hours and at the daily resolution it encompasses $4.8 \cdot 10^5$ days.

Figure 3a and b show the resulting precision for an hourly and daily resolution, respectively. Figure 3c and d show the relative contribution of every uncertainty term in Eq. 9, i.e. a high relative contribution indicates that the given parameter is an
important source for the overall uncertainty of SWE. Figure 3 shows that the main uncertainty can be attributed to the neutron count uncertainty, independently of the time resolution. However, the precision estimate presented here does not include the uncertainty of the correction parameterization (Eq. 2 and Eq. 3) or the conversion equation (Eq. 5) and its parameters (Table 4).

### 3.5 Estimating the uncertainty of automatically derived SD and snow density

In general, we distinguish between the observed standard deviation of all observed hourly values during one day ($s$) and the
theoretical measurement precision in those daily values ($\sigma$). The daily standard deviation of SWE ($s_{\mathrm{SWE\_crs}}$) and SD ($s_{\mathrm{SD\_sr}}$) are derived assuming a gaussian distribution. For the standard deviation of the bulk density ($s_{\rho\_(\mathrm{crs\_sr})}$), we apply gaussian error

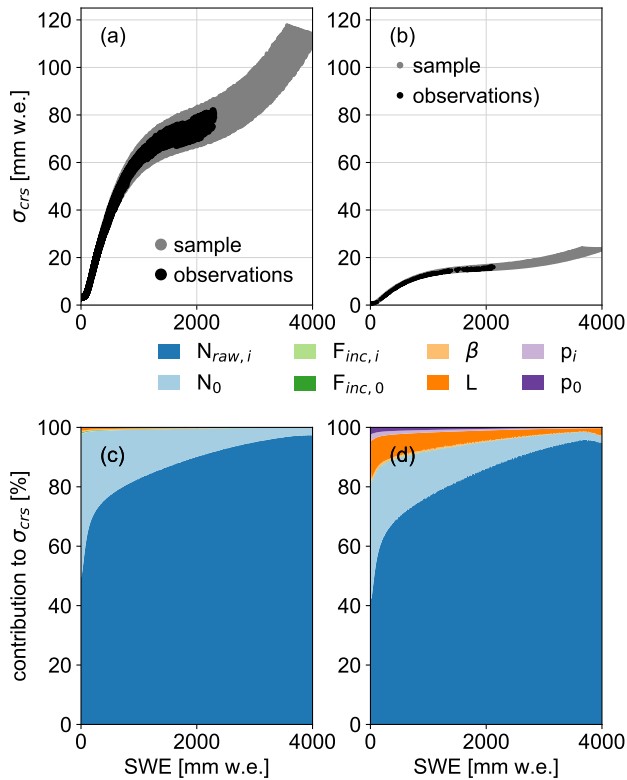

**Figure 3.** Precision of SWE calculated by means of error propagation. (a) and (b) show the absolute precision with grey dots as an synthetic data set and black dots as the in situ observations. (c) and (d) show the relative contribution of each parameter to the overall precision. (a) and (c) present the results based on hourly observations while (b) and (d) show the results of the daily observations.

propagation to Eq. 7 to yield

$$s_{\rho(\text{crs\_sr})} = \sqrt{\left(\frac{s_{\text{SWE\_crs}}}{SWE_{crs}}\right)^2 + \left(\frac{s_{\text{SD\_sr}}}{SD_{\text{sr}}}\right)^2} \tag{11}$$

The calculation of the measurement uncertainties of SD ($\sigma_{\text{sr}}$) and the bulk density ($\sigma_{\rho(\text{crs\_sr})}$) is described in the following paragraphs.

5    The uncertainty in daily SD observations varies with the depth of the snowpack. According to the installation manual, the accuracy lies between ±1 cm and 0.4% of the distance from sensor to ground (Campbell Scientific, 2016). Since the sensor is mounted at 4.8 m, the maximum uncertainty equals 1.9 cm under snow free conditions. In addition to the given uncertainty, we add a further systematic measurement uncertainty on SD's less than 30 cm. This uncertainty is caused by the footprint of the sonic ranging sensor which is large enough to include parts of the mast's foundations. The mast's foundation consists of three

10    wooden beams with a height of 20 cm each. They stabilize the mast on the glacier ice, especially during the ice melt season. To keep the wooden beams in place, they are anchored with tubes drilled into the ice. These tubes also exceed the 20 cm height of

the wooden beams and add an additional error. We estimate this additional uncertainty to be 30% with SD below 30 cm, 50% with SD below 25 cm, 80% with SD below 15 cm and 100% with SD below 10 cm. Moreover, the SD measurements from 20 January 2018 to 4 May 2018 which have been taken from another station at high-elevation carry an additional uncertainty of 6 cm (see Table 3).

Using the uncertainties of SD and SWE, we derive the uncertainty of the daily bulk density ($\sigma_\rho$) as

$$\sigma_{\rho(\text{crs\_sr})} = \sqrt{\left(\frac{\sigma_{\text{crs}}}{SWE_{\text{crs}}}\right)^2 + \left(\frac{\sigma_{\text{sr}}}{SD_{\text{sr}}}\right)^2} \tag{12}$$

## 3.6    Estimating the uncertainty of field data

Field measurements carry uncertainties from a variety of sources (sampling tube, weight scale, sampling technique, etc.). Few studies discuss the accuracy of SWE observations comprehensively (e.g. Stuefer et al., 2013). Commonly, a relative uncertainty

of $\pm$10% is applied (e.g. Schattan et al., 2017). Thibert et al. (2008), for example, focus on uncertainties for glacier mass balance calculations based on the glaciological method. These random and systematic errors, however, assume underlying firn with unknown water content and are not intended for snow accumulation. For our study, we have chosen to calculate an uncertainty based on the gaussian error propagation (see Papula, 2010). Next to the human-induced errors, which cannot be quantified in the scope of this study, we identify two major sources of sampling errors. These are related to the weighed mass

and the snow volume within the tube.

We sample an entire column of the snowpack from the surface to the snow-glacier interface. These samples are taken either within a snow pit or by extracting a snow core. In both approaches we use a sampling tube. In deeper snowpacks the whole column cannot be sampled in one measurement step. Thus we take several samples with a certain length ($l_{\text{s}}$) from snow to glacier surface. For each of these samples the density ($\rho_s$) is calculated by applying Eq. 13. The variable $m_{\text{s}}$ represents the

mass of the snow weighed in situ with a scale while $r_{\text{tube}}$ represents the radius of the sampling tube.

$$\rho_{\text{s}} = \frac{m_{\text{s}}}{\pi \cdot r_{\text{tube}}^2 \cdot l_{\text{s}}} \tag{13}$$

The sources of the sample uncertainty in density ($\sigma_{\rho\_s}$) arise from the uncertainties in snow-mass weighing ($\Sigma m_{\text{s}}$), the uncertainties of the sampled volume given by the radius ($\Sigma r_{\text{tube}}$) and the uncertainties of the sampled length ($\Sigma l_{\text{s}}$) in the snowpack. The uncertainty of the mass is thus composed of two individual sources; the scale for weighing the sample ($\Sigma m_{\text{scale}}$),

and the extracted snow volume ($\Sigma m_{\text{mass}}$). These two uncertainties are added to $\Sigma m_{\text{s}}$ following gaussian error propagation. Because the surface area of the extracted snow core does not always match the tube's surface area, we define an uncertainty range for the radius. The relative uncertainty of each sample ($\sigma_{\rho\_s}$) is then derived from

$$\sigma_{\rho\_s} = \frac{\Sigma \rho_s}{\rho_s} = \sqrt{\left(\frac{\Sigma m_{\text{s}}}{m_{\text{s}}}\right)^2 + \left(2 \cdot \frac{\Sigma r_{\text{tube}}}{r_{\text{tube}}}\right)^2 + \left(\frac{\Sigma l_{\text{s}}}{l_{\text{s}}}\right)^2} \tag{14}$$

Given the density for each sample at different depths within the snowpack, we calculate the bulk density ($\rho_{\text{field}}$). To this end,

we need to divide the snowpack into layers of variable lengths. Because of this variation, we determine a multiplicative weight

$p_{\mathrm{l}}$ for each layer as

$$p_{\mathrm{l}} = \frac{l_{\mathrm{l}}}{SD_{\mathrm{field}}} \tag{15}$$

This weight corresponds to the relative contribution of $l_{\mathrm{l}}$ to the total depth of the snowpack ($SD_{\mathrm{field}}$) which is measured independently.

The samples may overlap depending on the tube and the extraction method used. If there is no overlap, the length of the sample $l_{\mathrm{s}}$ is equal to the thickness of the layer $l_{\mathrm{l}}$, and the number of samples is equal to the number of layers ($n_{\mathrm{l}}$). Simultaneously, the sample density ($\rho_{\mathrm{s}} \pm \Sigma\rho_{\mathrm{s}}$) corresponds to the layer density ($\rho_{\mathrm{l}} \pm \Sigma\rho_{\mathrm{l}}$). If the samples overlap, $\rho_{\mathrm{l}}$ corresponds to the mean density and propagated uncertainty of the overlapping samples. In that case, the number of layers is greater than the number of samples. With

$$\rho_{\mathrm{field}} = \frac{1}{n_{\mathrm{l}}} \cdot \sum_{i=1}^{n_{\mathrm{l}}} p_{\mathrm{l},i} \cdot \rho_{\mathrm{l},i} \tag{16}$$

and

$$\sigma_{\rho\_\mathrm{field}} = \frac{\Sigma\rho_{\mathrm{field}}}{\rho_{\mathrm{field}}} = \frac{\sqrt{\sum_{i=1}^{n_{\mathrm{l}}} p_{\mathrm{l},i} \cdot \left(\Sigma\rho_{\mathrm{l},i}\right)^2}}{\rho_{\mathrm{field}}} \tag{17}$$

we obtain the bulk density ($\rho_{\mathrm{field}}$) and its relative uncertainty ($\sigma_{\rho\_\mathrm{field}}$).

     Knowing the bulk density and the depth of the snowpack ($SD_{\mathrm{field}}$), we calculate total SWE ($SWE_{\mathrm{field}}$) with

$$SWE_{\mathrm{field}} = \rho_{\mathrm{field}} \cdot SD_{\mathrm{field}} \tag{18}$$

With gaussian error propagation we derive the relative uncertainty of SWE as

$$\sigma_{\mathrm{SWE\_field}} = \frac{\Sigma SWE_{\mathrm{field}}}{SWE_{\mathrm{field}}} = \sqrt{\left(\frac{\Sigma\rho_{\mathrm{field}}}{\rho_{\mathrm{field}}}\right)^2 + \left(\frac{\Sigma SD_{\mathrm{field}}}{SD_{\mathrm{field}}}\right)^2} \tag{19}$$

The absolute uncertainty of $SD_{field}$ ($\Sigma SD_{\mathrm{field}}$) is estimated independently of the sample measurements. The absolute uncertainty of the bulk density ($\Sigma\rho_{\mathrm{field}}$) is given in Eq. 17.

For each field campaign we define the uncertainty based on the sampling tube, the scale, and whether we sampled within a snow pit or extracted a snow core. We used tubes with a radii of 4.00±0.10 cm, 4.15±0.15 cm, 4.50±0.10 cm and 4.75±0.10 cm and lengths of 117.0 cm, 107.0 cm, 55.7 cm and 56.0 cm, respectively. Additionally, we have three scales with a maximum weighing capacity of 2±0.02 kg, 5±0.05 kg and 12±0.10 kg. The uncertainty in the weighed mass ranges from 0.05 kg to 0.15 kg depending on the snow depth and the tube length. Sampling lengths are attributed an uncertainty

from 0.5 cm to 1.0 cm. During a campaign we usually sample more than one snow column. In those cases we take an average of all snow variables and average all uncertainties to yield the mean uncertainties. We quantify the variability within several snow columns with their standard deviation ($s$) which is smaller than the mean uncertainty. An extensive table on all assumed uncertainties, the number of samples per snow pit is provided in the supplement.

## 3.7 Pre-processing and precipitation scaling

In the final part of this study we estimated the optimal scaling factor for precipitation amounts from three stations at lower elevations and for three grid cells of the gridded precipitation (Table 2).

Because snow accumulation is cumulative and precipitation is instantaneous we first sum the hourly precipitation amounts to daily amounts over the whole winter seasons from HH:01 to (HH+1):00. Second, we adjust the cumulative precipitation to the amounts of snow accumulation at the beginning of the season. In the first winter season (2016/17), precipitation records begin at the same time as snow accumulation observations. In the second winter season (2017/18), observations by the CRS began when the snowpack was already developed. To start the observations of snow accumulation and cumulative precipitation at the same level, we add a constant offset to the cumulative precipitation. This offset corresponds to the first SWE amount measured by the CRS in the respective winter. The end of the precipitation time series is set at the end of May for both years. At this point in time, the peak of SWE had already passed in both winters.

In a first analysis we apply scaling factors between 0.1 and 8.1 at an 0.1 interval to all daily instantaneous precipitation observations. We then accumulate the scaled daily precipitation over the winter season. Compared to 392 days of CRS observations, we calculate the daily absolute error and derive the seasonal mean absolute error (MAE). We then chose the scaling factors resulting in the lowest MAE for all AWS and grid cells.

In a second analysis we find the optimal scaling factor for each precipitation phase, i.e. solid, liquid and mixed-phase. The precipitation phases are defined through air temperature observations at the glacier site. This parameterization of precipitation phases is based on values from literature. The study by Sims and Liu (2015), for instance, show that 90% of precipitation events were solid precipitation for near-surface temperatures below $0°C$ for land surface observations. For temperatures above $3°$, more than 85% of all precipitation events were liquid. Hence, we consider all precipitation as liquid if temperatures are above $3°C$ during at least six hours. If temperatures remain between $0°C$ and $3°C$ during at least six hours, we classify it as mixed-phase precipitation. Solid precipitation only occurs with subzero temperatures. For each of these phases, we apply the procedure described in the analysis above.

## 4 Results

### 4.1 Measured SD, SWE and snow density

With the CRS installed on Plaine Morte, SWE was measured during two subsequent winter seasons (2016/17 and 2017/18). These two winters were markedly different. The first winter received typical snowfall while the second winter experienced particularly heavy snowfall. During winter 2016/17, a maximum SD of 324 cm was reached on 2 May 2017 and a maximum SWE (1379 mm w.e.) on 18 May 2017. With these observations, the first winter season lies in the range of average mean specific winter mass balances between 2009 to 2019 (GLAMOS, 1881-2018). During the second winter (2017/18), a maximum of 527 cm of SD (1 April 2018) and a maximum SWE of 2122 mm w.e. (24 May 2018) were observed, which corresponds to approximately 1.5 times of the SWE amount of the previous year.

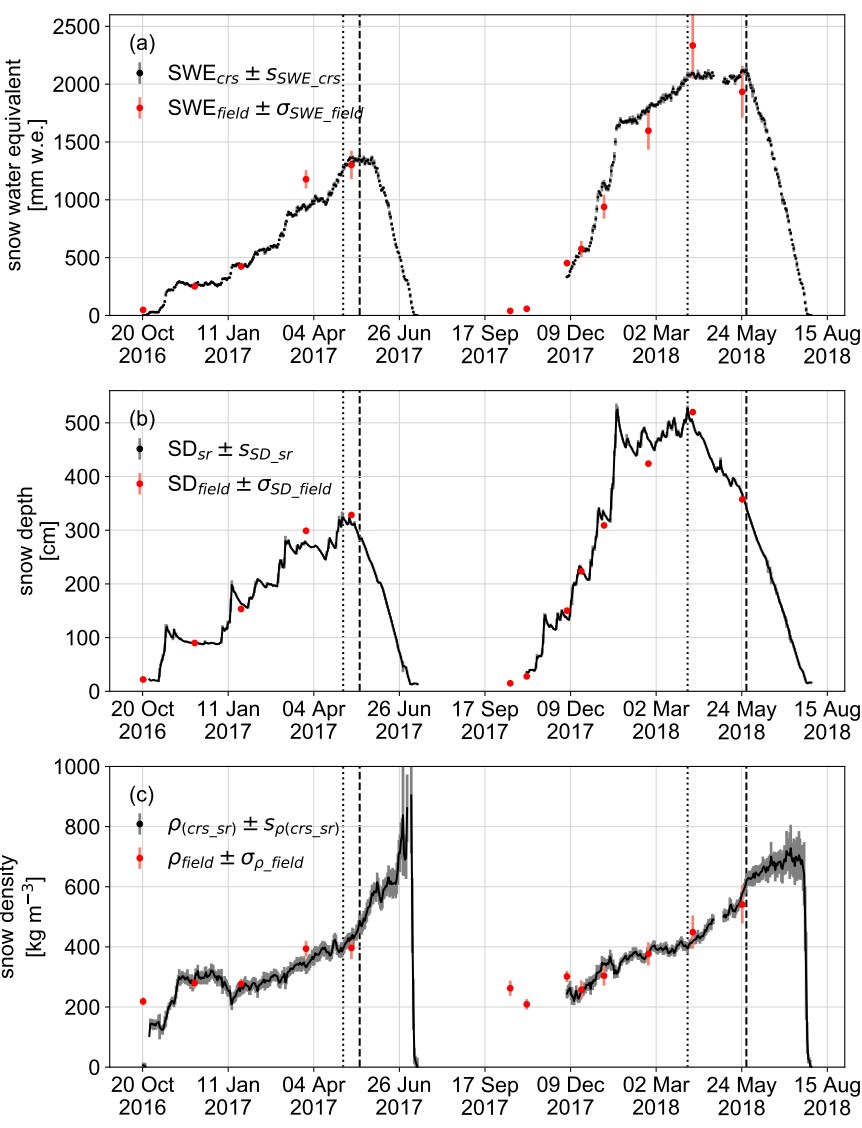

**Figure 4.** Continuous observations of (a) SWE, (b) SD and (c) snow density with their daily standard deviation. The red dots show the manual field measurements with their uncertainties (salmon bars). The dotted (dashed) line shows the day of the seasonal maxima in SD (SWE).

Figure 4a and b show the snow accumulation and ablation over the two winter seasons. In both winters, the first snowfall occurred mid-October when SDs reached about 20 cm. By mid-November, SD exceeded one meter with a SWE amount of approximately 300 mm w.e. in both winters. In winter 2016/17, the SD remained almost constant until the beginning of January 2017. In the following winter, SD significantly increased from November 2017 to January 2018. By that time, it had already surpassed the maximum in SD of the previous winter.

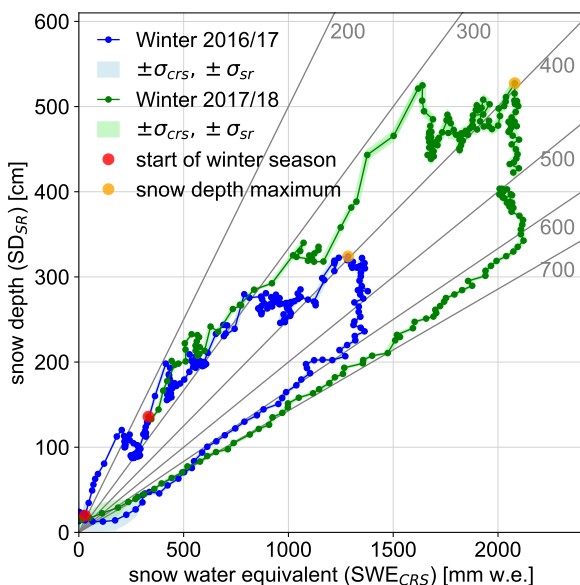

**Figure 5.** Daily mean SWE and daily mean SD. Grey lines show the densities in $kg\,m^{-3}$. The beginning of each winter season is marked by the red dot, yellow dots indicate the maximum of SD.

From January 2017 to May 2017, SD increased almost continuously with a period of accumulation followed by a period of densification. The time lag between the maximum of SD and the maximum of SWE is 16 days. During this time span, SWE remained almost constant and only increased little. By the beginning of July 2017, the snow had completely melted. In winter 2017/18, SWE increased more continuously between end of January and beginning of June. In this winter, the maxima in SD

5   and SWE are almost two months apart. Already in April 2018, SD started decreasing while SWE remained constant. During that time, only few events led to small increases in SD. From end of Mai 2018 onwards, SWE decreased rapidly. By the end of July, the snow had disappeared.

In the beginning of winter 2016/17 snow density increases after a short decrease (Fig. 4c). This short increase corresponds to the snowfall observed in Fig. 4a and b. In general, densification slowly progresses with short intervals of decreasing densities

10   caused by snowfall. Between the maximum SD and the maximum SWE, density increases almost linearly in both winters. After reaching the maxima of SWE, snow densities are above $480\,kg\,m^{-3}$. Shortly before the snowpack disappears completely, densities decrease rapidly. The comparatively high standard deviations of the snow density (Fig. 4c) are a consequence of the daily variability in SD. SD may decrease significantly during a day when densification rates are strong while SWE remains constant.

15   Figure 5 shows daily SWE in relation to daily SD over the winter season. During the accumulation period, daily densities vary between $200\,kg\,m^{-3}$ and $400\,kg\,m^{-3}$. An increase in SD is often followed by a period of constant SWE and decreasing SD which is characteristic of snow densification. Both winters tend to follow a similar pattern in the evolution of density. At

**Table 6.** Overview of all considered processes, their criteria and the number of days when the criteria are fulfilled. The colors refer to the processes displayed in Fig. 7b.

| Process | color | snow depth | SWE | days (2016-2018) | days (2016/17) | days (2017/18) |
|---|---|---|---|---|---|---|
| all | - | - | - | 487 (100%) | 260 (100%) | 227 (100%) |
| accumulation | lightblue | $> \sigma_{sr}$ | $> 0$ cm | 107 (22%) | 56 (22%) | 51 (22%) |
| ablation | red | $< -\sigma_{sr}$ | $< -\sigma_{crs}$ | 110 (23%) | 48 (18%) | 62 (27%) |
| densification | green | $< -\sigma_{sr}$ | $\geq 0$ cm | 174 (36%) | 80 (31%) | 94 (41%) |
| not classified | white | - | - | 96 (20%) | 76 (29%) | 20 (9%) |
| subgroups | | | | | | |
| accumulation (high confidence) | blue | $> \sigma_{sr}$ | $> \sigma_{crs}$ | 81 | 47 | 34 |
| densification with accumulation | lightgreen | $\leq 0 \, cm$ | $> \sigma_{crs}$ | 74 | 43 | 31 |

the maximum of SD, the daily density is $390 \, \mathrm{kg \, m^{-3}}$ for winter 2016/17 (2 May 2017) and $392 \, \mathrm{kg \, m^{-3}}$ for winter 2017/18 (1 April 2018). After these peaks, the snowpack begins to densify continuously. During this period of densification, SWE remains almost constant while SD decreases by about 1 m (2016/17) and 1.5 m (2017/18). Only then does SWE begin to decrease simultaneously with SD, following the density lines between $600 \, \mathrm{kg \, m^{-3}}$ and $700 \, \mathrm{kg \, m^{-3}}$.

5     For the evaluation of the CRS, we use field data from nine campaigns over the two winter seasons. Figure 6 shows the autonomous data of SWE (Fig. 6a), SD (Fig. 6b) and snow density (Fig. 6c) compared to the data from the field surveys. On average, the CRS overestimates SWE by +2%±13%. The SD measurements agree within a standard deviation of ±6%, and ±7% when also considering the interpolated data during the measurement gap of SD. The snow density data agree on average with a standard deviation of ±9% (Fig. 6d). The correlation coefficients ($r^2$) of all considered snowpack parameters are higher 10  than 0.89 (Fig. 6).

## 4.2   Daily variations of SD and SWE

With the continuous data of SWE and SD, we evaluate the daily variations of snow properties (Fig. 7a). We use this to classify days based on whether they were dominated by accumulation, densification or ablation. For this purpose, we define criteria for SWE and SD considering the precision estimates of the observations. The precision is especially important for the SWE 15  measurements because we want to distinguish between noise and signal. Table 6 gives an overview of all criteria and the number of days when these are fulfilled. A day dominated by accumulation has to have a change in SD greater $\sigma_{sr}$, while the change in SWE has to be greater than 0 cm. To ensure more confidence (accumulation with high confidence), the SWE changes have to exceed $\sigma_{crs}$ (Table 6). The same applies for ablation with the difference that the daily change has to be more negative than the uncertainty values. For densification, we require a significant decrease in SD while SWE remains constant 20  or increases. For the latter case, we extract the days where densification and accumulation occur on the same day. Figure 7b shows that the winter is mainly dominated by accumulation and densification. Some days with ablation occur at the beginning

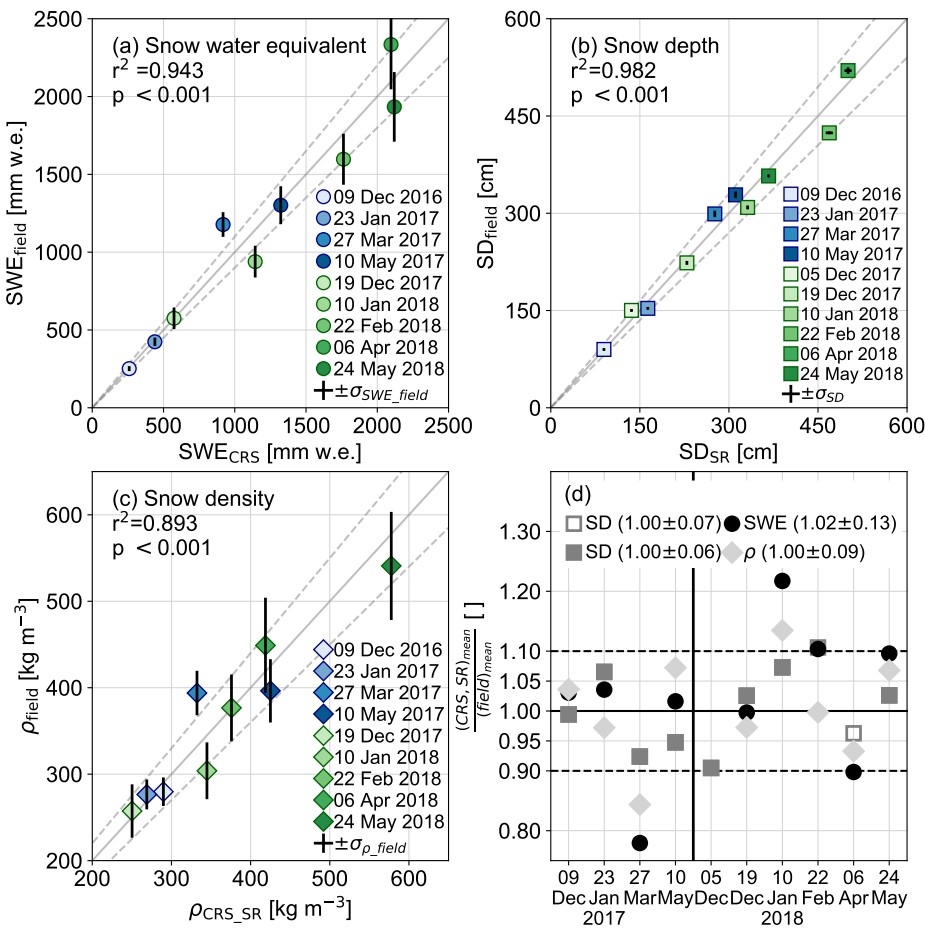

**Figure 6.** Scatter plots of the field data compared to the autonomous measurements of (a) SWE, (b) SD and (c) bulk snow density. The dashed grey lines show the range within ±10%. The error bars correspond to the uncertainty of the field data only for SWE and snow density. (d) shows the ratio between field and automatic measurements for SD, SWE and snow density ($\rho$). The unfilled grey squares represent the data interpolated from another station.

of March 2017. On these days, we note higher wind gusts and subzero temperatures (Fig. 7c and d). Hence, ablation is likely caused by snow drift rather than melt.

The mean daily meteorological conditions can be summarized for the categorized days (Fig. 8, Table 6). Days with accumu-
lation are characterized by high relative humidity (Fig. 8a), significantly lower temperatures (Fig. 8b) and an average decrease
5   in mean density (Fig. 8c). Wind speeds are often above $6\,\mathrm{m\,s^{-1}}$ and originate mainly from south over west to north. Days dom-
inated by ablation are characterized by average daily relative humidity (Fig. 8a), significantly higher temperatures (Fig. 8b)
and wind speeds that are mainly around or below $4\,\mathrm{m\,s^{-1}}$ (Fig. 8d). During ablation days, we find no significant change in
density. Days with densification are drier than days with ablation. The median values of daily mean temperatures are similar to

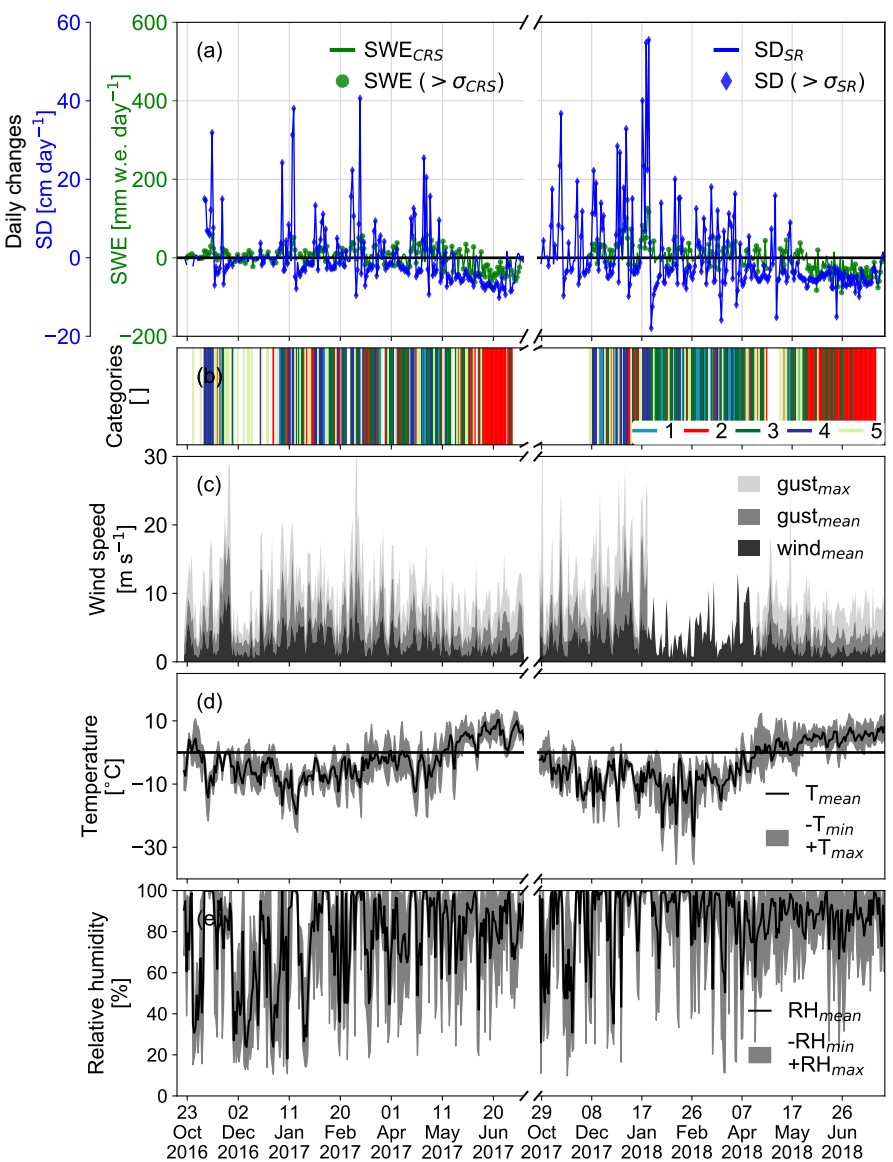

**Figure 7.** (a) Daily changes of SWE (green) and SD (blue). (b) Categorization of process-dominated days with 1 as accumulation, 2 as ablation, 3 as densification, 4 as accumulation with high confidence, and 5 as densification with accumulation (see Table 6). (c) Daily mean wind speeds (black shading), mean wind gusts (grey shading) and maximum wind gust (light grey shading). (d) Daily mean temperature (black line) with maximum and minimum temperature (grey shading). (e) Daily mean relative humidity (black line) with daily minimum and maximum relative humidity (grey shading).

the ones in the reference periods. Winds originating from the sectors southwest to south are usually below 6 m s$^{-1}$ during days with densification. More frequently, however, the wind blows from south-east and is rather strong during the ablation days.

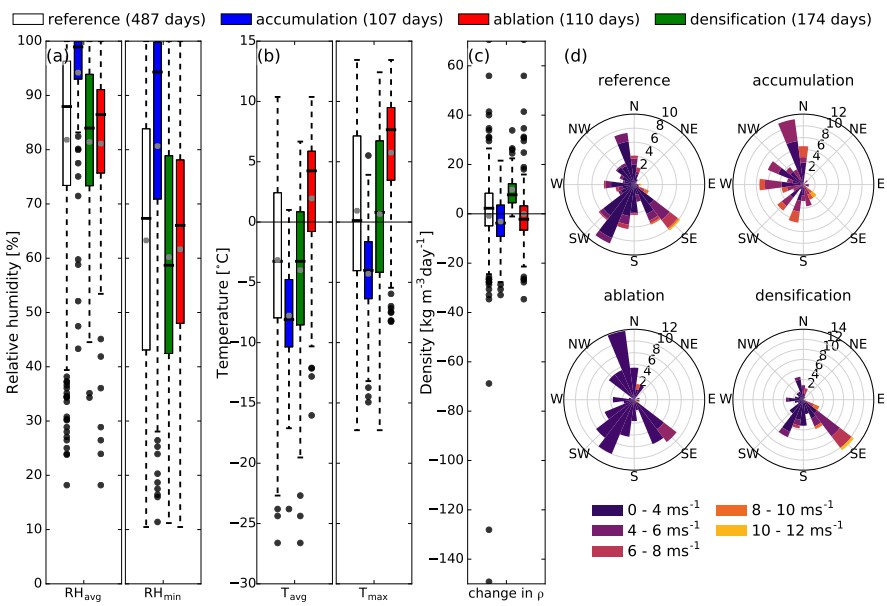

**Figure 8.** Summary of meteorological conditions during process-dominated days (accumulation, ablation and densification). (a) Daily mean and minimum relative humidity, (b) daily mean and maximum temperature, (c) the change in mean bulk snow density, and (d) mean daily wind speeds and -direction. The numbers in the wind roses correspond to the percentage of days within that selection. The reference includes all days with valid data (487 days).

All these findings align with our general expectations that accumulation occurs with lower temperatures, high relative humidity, and stronger winds. Ablation through melt is mainly characterized by higher temperatures, lower relative humidity and lower wind speeds. Of all densification days, 43% show a simultaneous increase in SWE ("densification with accumulation", Table 6). When both processes occur at the same day, it suggests either simultaneous compaction of snowfall, accumulation by snow drift, or infiltration of liquid precipitation. About 40% of these days have negative temperatures and low wind speeds, while the remaining 60% have either positive temperatures or wind speeds above 4 m s$^{-1}$. Positive daily temperature might suggest infiltration within the snowpack. Higher wind speeds would rather suggest an effect of snow drift.

### 4.3 Precipitation scaling

With the daily observations of SWE we assess the accuracy of an approach utilizing scaled precipitation from three nearby AWS and RhiresD. In the following, we refer to the autonomous CRS measurements as snow accumulation or SWE.

Without applying a scaling factor, we see a large difference between cumulative precipitation and snow accumulation on the glacier (Fig. 9). This could be due to the high spatial variability of solid precipitation and/or undercatch of rain gauges (Kochendorfer et al., 2017; Pollock et al., 2018).

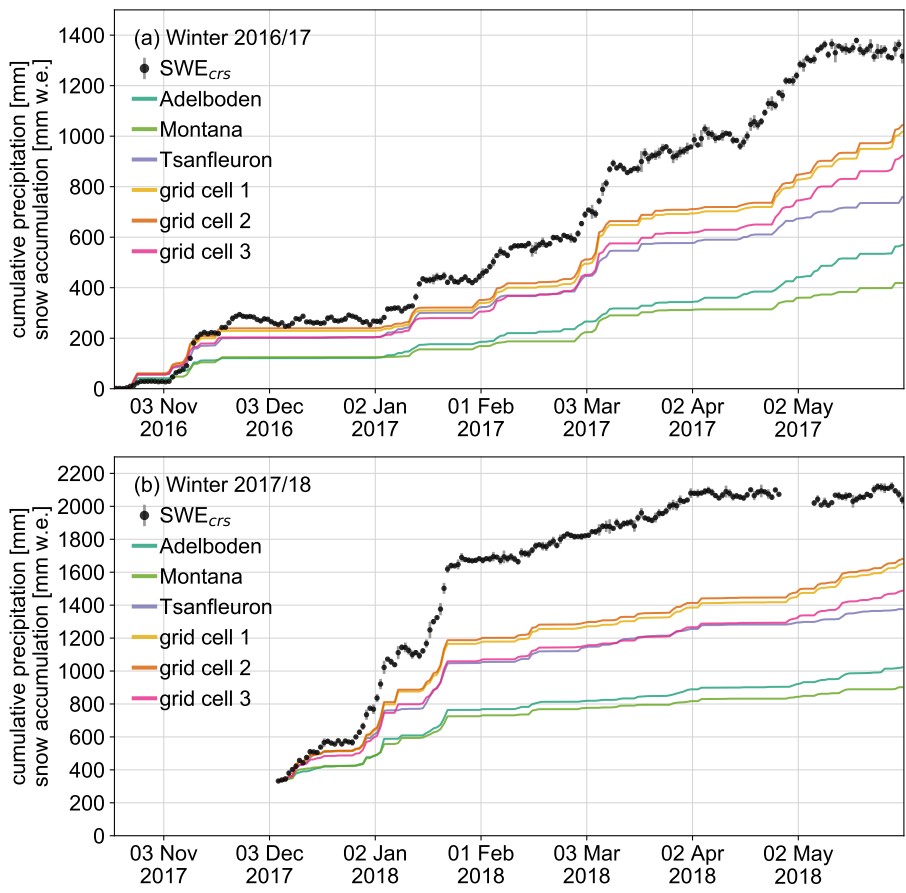

**Figure 9.** Cumulative precipitation and snow accumulation of (a) winter 2016/17 and (b) winter 2017/18. Black dots show daily SWE observations with their standard deviations. Colored lines represents all cumulative precipitation time series.

In general, the onset of snow accumulation corresponds well with increases in the precipitation observations. However, in November/ beginning of December 2016 increases in SWE are observed with no corresponding increase in precipitation registered at the AWS (Fig. 9a). In addition, Montana captures fewer precipitation events than other stations in winter 2016/17 (Fig. 9a). In winter 2017/18, Montana captures approximately the same number of events as the other AWS (Fig. 9b). At times

5   when SWE increases without an increase in precipitation (e.g. December 2016, mid-February 2017 and mid-February 2018) we also observe higher mean daily wind speeds and maximal wind gusts (Fig. 7c). This suggests snow drift as an explanatory mechanism. Precipitation amounts are generally lower than SWE amounts observed for individual events on the glacier. This bias seems to increase with the duration of the precipitation events, especially for those lasting several days.

The optimal scaling factors range from 1.4 (grid cell 2) to 3.3 (Montana). Gridded precipitation and Tsanfleuron have similar

10   scaling factors (Table 7). The two AWS have significantly higher scaling factors with 2.8 (Adelboden) and 3.3 (Montana). The best performance is found for Tsanfleuron with a MAE of 70±37 mm w.e.

**Table 7.** Scaling factors resulting in a minimal MAE. Two different approaches are presented; one factor applied and three factors applied distinguishing between the precipitation regimes (solid, mixed and liquid).

| | one factor | | three factors | | | |
| | all [ ] | MAE [mm w.e.] | solid [ ] | mixed [ ] | liquid [ ] | MAE [mm w.e.] |
|---|---|---|---|---|---|---|
| Adelboden | 2.8 | 79±55 | 2.9 | 1.3 | 0.6 | 41±37 |
| Montana | 3.3 | 76±42 | 3.4 | 2.5 | 0.8 | 56±37 |
| Tsanfleuron | 1.8 | 70±37 | 1.8 | 1.4 | 0.5 | 50±29 |
| grid cell 1 | 1.5 | 72±43 | 1.5 | 0.9 | 0.3 | 48±43 |
| grid cell 2 | 1.4 | 74±59 | 1.5 | 0.8 | 0.3 | 39±30 |
| grid cell3 | 1.7 | 78±43 | 1.7 | 0.9 | 0.3 | 46±44 |

Accumulation events in October/ November 2016 and May 2017 are represented fairly poor by precipitation observed at AWS. Accumulation is overestimated by at least 50 mm w.e. (Fig. 10a and b). In these months, hourly temperatures at Plaine Morte reach values higher than 0°C (Fig. 10e). It is thus likely that precipitation falls as rain rather than snow. Nonetheless, it may still contribute to SWE by refreezing.

The temperature-dependent parameterization for the precipitation phases result in potentially 68 days (17%) with liquid precipitation, 288 days (72%) with solid precipitation, and 46 days (11%) with mixed-phase precipitation. Table 7 provides the resulting optimal scaling factors for each of these phases. The scaling factors for solid precipitation remains similar to the first analysis (Fig. 10a). Mixed-phase and liquid precipitation are scaled by lower factors. Scaling factors for liquid precipitation are smaller than a factor of one (Table 7). With these scaling factors, we reduce the MAE to below 60 mm w.e. and the temporal evolution is generally consistent with the SWE observations on the glacier (Fig. 10c).

## 5 Discussion

### 5.1 CRS performance and limitations

The CRS shows a good agreement with manual field measurements. On average, the CRS overestimates SWE by +2%±13%. The agreement of the individual field campaigns varies between an excellent agreement within ±2% (10 May 2017 and 19 December 2017) and a rather large difference of more than ±20% (27 March 2017 and 10 January 2018). Otherwise, the agreement is within the uncertainty of the manual field measurements.

In the second winter season, SWE amounts were exceptionally high with more than 2000 mm w.e. Nevertheless, the agreement to field measurement is within ±10% indicating that the measurement limit of SWE has not yet been reached. Due to the exponential nature of the relationship there is no distinct threshold beyond which the relative neutron count is no longer sensitive to SWE (Fig. 2).

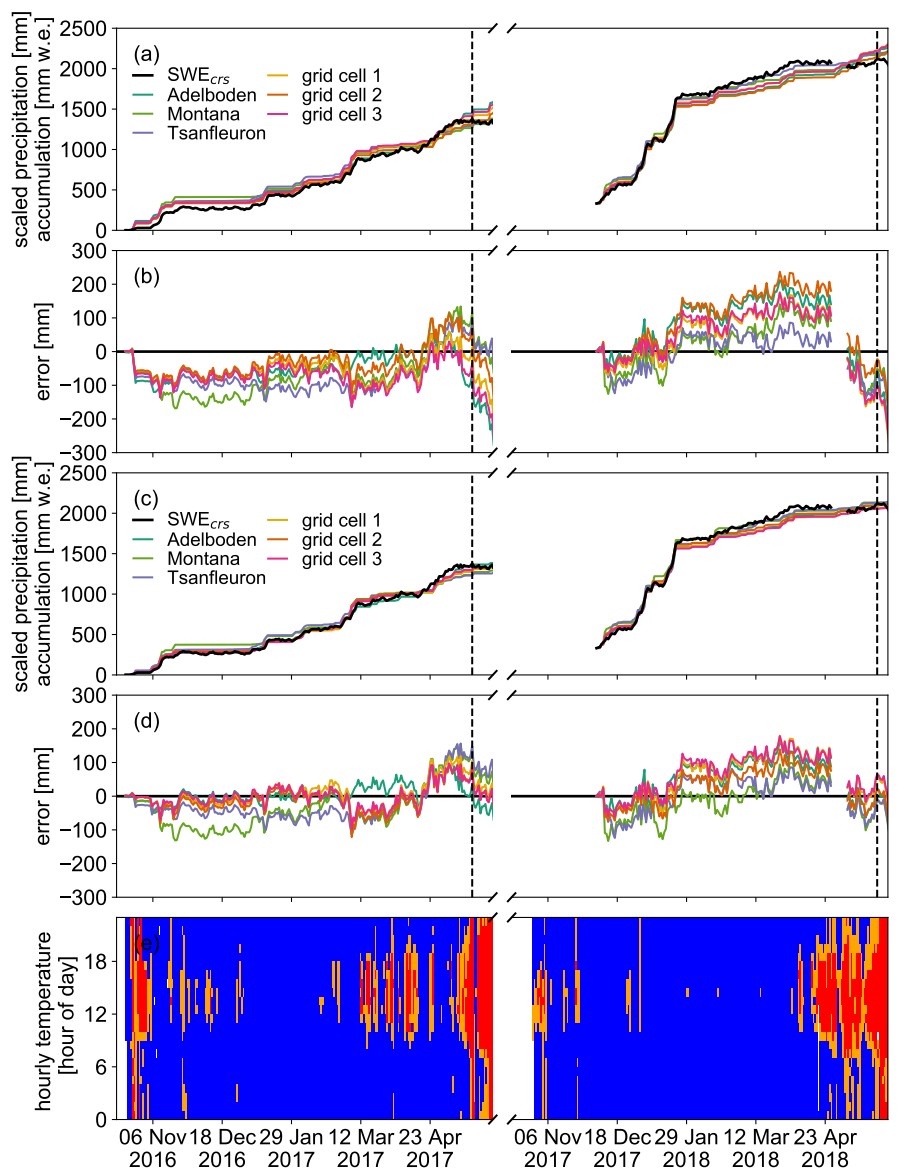

**Figure 10.** SWE observations (black dots) and scaled cumulative precipitation with (a) one factor and (c) three precipitation phase-dependent scaling factors (see Table 7). (b) and (d) show the difference between precipitation and SWE of (a) and (c), respectively. The hourly temperature at Plaine Morte is visualized in (e) with temperatures above 3°C colored in red, between 0°C and 3°C in orange, and below 0°C in blue. The dashed line corresponds to the date of the seasonal maximum in SWE.

The data processing of the neutron counts as presented is straightforward. Given the transformation equation, only the initial neutron count rate can be calibrated. But a variation of this calibration parameter within its uncertainties has little influence on the resulting SWE amounts, especially for amounts larger 400 mm w.e. This is a consequence of the exponential nature of the

conversion equation (Eq. 5). More importantly, the correction of the raw neutron count rate with solar activity and air pressure may influence the results. The applied correction functions have been previously used for SWE (e.g. Howat et al., 2018) or soil moisture studies (e.g. Zreda et al., 2012; Andreasen et al., 2017). In contrast to previous studies of above-ground CRS, changes in atmospheric moisture are not taken into consideration. We assume that for the sub-snow CRS, fast neutrons are produced

within the snowpack rather than in the atmosphere, an assumption also made in Howat et al. (2018), and implicitly made by preceding authors in their studies (e.g. Kodama et al., 1979; Paquet and Laval, 2005; Gottardi et al., 2013). Another source of uncertainty is the semi-empirical fit that has been used in this study. Because this study focuses on the application for snow and glacier studies, we applied the relations used by Howat et al. (2018). In general, the conversion function has the potential to introduce considerable uncertainty in the inferred SWE. However, the applied empirical relation has shown to be adequate

as the resulting SWE agrees well with independent field measurements, indicating only a minor bias and a standard deviation for individual observations that lie in the range of the uncertainty of the in situ SWE surveys.

For all correction factors such as air pressure and solar activity, an estimated uncertainty was propagated through all equations and showed that the resulting precision is mainly defined over the neutron count rate (Fig. 3c and d). Assuming that the parameterization of the correction equations carry no uncertainties, the influences of all other measurements and constant

parameters are small. Moreover, an independent study by Howat et al. (2018) quantified a precision of 0.7% of a CRS lying below the snowpack on the ice sheet. Their precision is based on the daily standard deviation of hourly observations and does not include uncertainties of the correction factors or the conversion equation. Moreover, their results are affected by lower in situ air pressure with consequently higher neutron count rate in addition to lower SWE amounts. The latter places them on a steeper part of the calibration curve (Fig. 2) where changes in neutron counts are more sensitive and have a higher precision.

In general, the precision can be increased by integrating over longer time periods.

The main advantage of the CRS is that it can be deployed in an exceptionally wide variety of terrain. There is no need for a stable and flat surface nor does it depend on the reception of satellite signal for its measurements (cf. Section 1.1). Air pressure which is needed for correcting the raw neutron count rate can either be interpolated from a nearby station or measured in situ without the need of an elaborate measurement setup. However, these advantages only apply for SWE measurements. As soon

as further observations such as SD or other meteorological parameters are required, the potential deployment areas become more limited.

## 5.2 Evolution of snow density

The snowpack of the two presented winters evolved differently in terms of amounts and accumulation rates. However, the evolution of the mean density of the snowpack is similar between the two winter seasons. The evolution before the onset of

melt agrees well with the findings of Mizukami and Perica (2008) and Saito et al. (2012). But snow densities become quite high ($>600\,\mathrm{kg\,m^{-3}}$) during the melt season in our study. After the SD maxima snow densities did not exceed $500\,\mathrm{kg\,m^{-3}}$ in Saito et al. (2012). A study in the Austrian Alps by Schattan et al. (2017) also shows lower mean snow densities towards the end of the snowpack. The high snow densities presented here could be a result of changes in the snow properties, measurement errors of SWE and SD estimations (Eq. 7), or a combination of both. Physical changes within the snowpack could be due

to refreezing of liquid water, water saturated snow in the top layers, locally thick ice lenses, or accumulation of liquid water around the CRS.

SWE from the CRS could, for example, be affected by a supraficial pond in the vicinity of the site. It remains unclear how such a hydrogen pool would influence the in situ point measurements of the sub-snow CRS. Other influences could come from the correction factors of the neutron count rate or the conversion equation applied in this study (cf. Section 5.1).

The SD measurements are also susceptible to errors. For example, the snow area below the sonic ranging sensor may show a small depression because of wind turbulence caused by the mast. Additionally, the snow around the main pole of the station melts faster possibly leading to a depression. It remains difficult to assess whether the radius of this depression would be within the footprint of the sonic ranging sensor. In winter 2017/18, the solar panels were submerged below the snow. To ensure further power supply, we had to dig them out. This snow pit around the main pole would have been refilled by wind, but densities are different, probably causing accelerated melt rates around the mast. For more shallow snowpacks, the metal anchorage of the mast's foundations might interfere with the SD measurements. The SD measurements, for instance, never observe a SD of 0 cm even though the sensor is calibrated for the mounted height and agrees well with the manual snow probings (Fig. 4b and Fig. 6).

In summary, this study setup shows that we are able to gain important information concerning the temporal evolution of snow density. We are able to derive the main periods of snow accumulation, densification and melt and it seems to follow a consistent pattern over two winter seasons (Fig. 5). In that sense, the CRS does not distinguish between water, snow and ice which avoids a falsification of SWE estimates. But it also becomes impossible to determine the snow layering. The SD observations seem to be more sensitive during the melting phase, probably because of the small-scaled heterogeneity in snow melt.

## 5.3 Estimating snow accumulation by precipitation scaling

The comparison of snow accumulation to precipitation observations is not without caveats given that snow accumulation is cumulative while precipitation is instantaneous. Moreover, snow accumulation is influenced by precipitation, snow drift and evaporation whereas precipitation is not. When continuous SWE measurements are unavailable, a straightforward and simple approach is to use precipitation data scaled to a ground reference. In the case of the glacier-wide mass balance studies on Plaine Morte this approach is applied using precipitation observations from Montana (GLAMOS, 2018). Precipitation data from AWS or RhiresD is freely available and highly resolved which makes it widely applicable. A drawback for AWS stations is the potentially large undercatch of solid precipitation combined with high wind speeds which can be on the order of a factor of three given solid precipitation and high wind speeds (Kochendorfer et al., 2017). The gridded precipitation, RhiresD, is potentially influenced by measurement errors as well as an underrepresentation of observations at high elevations. In addition, the complex topography of mountainous terrain is typically not sufficiently resolved in gridded data products.

The seasonal evolution of both winters could be reproduced with a constant factor between 1.4 and 3.3 for the AWS and RhiresD (Fig. 10 and Table 7). The minimal MAE is below 80 mm w.e. with larger absolute discrepancies during the second

winter season (Fig. 10b). In practice only a snapshot of SWE is available to scale precipitation data and thus the error at the daily resolution is most likely higher.

Applying a precipitation-phase-dependent scaling factor reduces the MAE to below 60 mm w.e. The phase of precipitation is parameterized using air temperature at the glacier site. Since air temperature is not as spatially heterogeneous as precipitation
it can be interpolated with less uncertainties. The parameterization proposed here distinguishes the precipitation phase between three temperature thresholds: below 0 °C, between 0 °C and 3 °C and above 3 °C. It is in line with previous studies. Jennings et al. (2018), for example, determined a snow-rain threshold between –0.4°C and 2.4 °C in the Northern Hemisphere.

The scaling factor for the solid phase remains similar to the overall constant factor because most precipitation falls in its solid form during the winter season. Mixed-phase precipitation is scaled with lower factors while liquid precipitation has scaling
factors below one. Especially for liquid precipitation, we observe a seasonal component. Liquid precipitation occurs mainly at the beginning and ending of the winter season (Fig. 10e). For winter 2016/17, for instance, the first precipitation event does not result in accumulated snow and therefore the constant scaling factor overestimates the beginning (Fig. 10b). At the end of the winter seasons, the snowpack is around its maximum and liquid precipitation would infiltrate the snowpack and refreeze contributing to an increase in SWE. To avoid liquid and mixed-phase precipitation, the time period in which precipitation is
accumulated could be adjusted. However, an adjustment of the time period would only partly exclude such events.

The choice of the precipitation data and AWS is also important. RhiresD has shown a better performance especially for the phase-dependent scaling factors. Tsanfleuron (2052 m a.s.l.) has the lowest constant factor (1.8) and MAE (70±37 mm w.e., Table 7) for the phase-independent approach. Adelboden and Montana which are located north and south of Plaine Morte have higher scaling factors than Tsanfleuron. In addition, they are on either side of the Alpine divide and dominated by different
weather regimes which is also confirmed by analyzing the temporal evolution. In winter 2016/17, many events captured by Adelboden are not represented in Montana (Fig. 9a). Nonetheless, Montana does not perform worse than Adelboden with only one constant factor. In the case of the phase-dependent scaling, the performance of Adelboden is significantly improved reducing its MAE by almost a factor of two.

Our calculation was possible only because we had reliable and continuous snow accumulation data. Due to the low spatial
variability of snow accumulation on Plaine Morte, the analysis can be made with a point measurement as a reference. But at high mountain sites with more topographic gradients, the location of the in situ measurement becomes more important which is why a glacier-wide mean is typically used. Another caveat of this assessment is the uncertainty of the CRS measurements which has not been taken into consideration. Nonetheless, the resulting MAE lie within ±13% of the average agreement between CRS and within the uncertainty of manual measurements.
In summary, it is possible to infer the temporal dynamics of snow accumulation at a high-elevation site by means of scaled precipitation data. However, at least one in situ observation is required for applying this approach. The choice of the precipitation data series and the time period considered is crucial for this methodology.

## 6 Conclusions and perspectives

During two winter seasons, we observed snow accumulation and ablation on a Swiss glacier at a daily resolution. The deployed CRS withstood the harsh environmental conditions at the high mountain site and measured reliably. The validation with manual field measurements indicated a mean accuracy of +2±13%. In combination with continuous SD measurements, the CRS provided daily mean snow densities that were within a range of ±8% of manual in situ snow density surveys.

With the daily mean snow density observations, we showed that the evolution of the bulk snow density can be divided into three main periods; accumulation, densification and ablation. Throughout the accumulation period, snow densities are low with periodical repetitions of snowfall and subsequent densification. At the seasonal maximum of SWE the snowpack densifies during several days before its melting period begins. Additionally, we investigated these three processes at a daily basis and could attribute general meteorological conditions to each process.

The deployment of the CRS on Plaine Morte provided continuous observations of SWE that could be used to assess the optimal scaling factor for readily available precipitation data. With the optimal scaling factor, we were able to obtain snow accumulation with a MAE of below 80 mm w.e. However, the performance depends on the choice of precipitation data, the choice of AWS, the date of the manual ground measurement and the time period considered. Scaling precipitation with a phase-dependent factor further improves these results.

In summary, we conclude that the CRS is a highly promising device for observing SWE continuously in cryospheric high alpine environments. Despite its limitations through the level of noise and its precision depending on absolute snow amounts, it is suitable for long-term monitoring of SWE in high mountain regions as well as polar regions. In such areas, its resilience in harsh environmental conditions, its rare need for maintenance (once it is properly running) and its flexibility regarding site topography are convincing. For shallower snowpacks, the temporal resolution can be increased to a sub-daily scale. For this study, we chose an elaborate measurement setup which would not be necessary if only SWE measurements are required.

In future, the point-scale footprint of the CRS should be better investigated by modelling of neutron trajectories. It would be particularly important to better quantify the influence of hydrogen pools in close vicinity of a sub-snow CRS. More investigations into the location-dependent correction of the solar activity would provide further insights into the applied processing of raw neutron counts. The deployment of additional CRS observations in other high-mountain regions of the Alps would not only give further indications on the suitability of precipitation scaling but also the spatial variability of snow accumulation.

*Data availability.* All observations at the Glacier de la Plaine Morte are available upon request from the first author. In future, it will also be available in an online repository.

*Author contributions.* RG prepared the manuscript, performed field work and data analysis with contributions from all co-authors. NS and MH contributed to the design and execution of the study. DD gave essential inputs on how to process the output of the cosmic ray sensor and define its measurement uncertainty.

*Competing interests.* Authors RG, NS and MH declare that they have no competing interests. Author DD is the owner of Hydroinnova LLC.

*Acknowledgements.* This study is supported by the Swiss National Science Foundation (SNSF), grant 200021_178963. MeteoSwiss provided the data of all stations besides the one at Plaine Morte. Jungfraujoch neutron monitor data were kindly provided by the Cosmic Ray Group, Physikalisches Institut, University of Bern, Switzerland. We like to thank H. Gubler from ALPUG, W. Jäger from WALJAG and T. Sarbach from Sarbach Mechanik GmbH. They designed the mast, programmed the logger box, and helped with the field installation. Furthermore, we like to acknowledge all the field helpers during the two winter seasons. Without them, this study would not have been possible. Last but not least, we thank the two anonymous reviewers and the editor for their constructive feedback and suggestions that significantly improved the manuscript.

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
