# Peer review of "Continuous and autonomous snow water equivalent measurements by a cosmic ray sensor on an Alpine glacier"

_The Cryosphere, 2019_

## Referee Comment (RC1) · Anonymous Referee #1 · 28 Jun 2019

General comments

This paper presents the application of a sub-merged cosmic ray sensor (CRS) on a Swiss glacier to derive daily snow water equivalent (SWE) values for two winter seasons. An additionally installed snow depth (SD) sensor was used to calculate the snow density by CRS SWE and SD. For validation, some manual field measurements were conducted within the two years and precipitation recordings from nearby weather stations as well as a gridded precipitation product were scaled to compare them with the measured CRS SWE. The measurement results derived by CRS are very plausible for snow accumulation, densification and ablation phases. In general, this paper is well

written and is, in my opinion, a good contribution to this journal. All measurements are well described and indicated by potential uncertainties and illustrated by significant figures. However, the main focus/objective of this paper has to be better defined. Are you rather interested in gaining better snow density information or are you mainly focusing on using and validating CRS measurements especially on such a glacial test site, or both? Please emphasize on this – maybe also the title has to be changed accordingly. In some paragraphs, references should be added or revised. Below, I indicated some other moderate to minor issues.

Specific comments

Please use the same units throughout the paper. SWE is usually given in mm (or kg/m$^2$), not in cm w.e.

Abstract

- p.1, l.12-16: The aspect of the comparison of the cosmic ray SWE values and the scaled precipitation is represented quite dominant in the abstract. I think this aspect can be reduced to two 2 sentences and the abstract should better include also a statement on the general applicability.

1. Introduction

- p.2, l.2: Not only the cold and windy conditions are a big challenge for in situ snow measurements in high mountains; please add that they are also often limited by difficult accessibility, complex terrain etc.

- p.2, l.13-21: The statements in this paragraph should be revised carefully as some statements are not correct. Some explanations are given in the following: Schmid et al. (2014) combined the upGPR with a snow depth sensor to additionally derive the liquid water content in snow. The main reasons for combining upGPR travel time with GPS signal strength in Schmid et al. (2015) were to eliminate an overestimation in snow depth during wet snow conditions, which would be the case by using only upGPR

measurements, and to be independent of poles as both sensors were buried beneath the snowpack, which could be useful, e.g., in avalanche prone slopes. Moreover, with this upGPR-GPS sensor combination it was possible for the first time to derive SWE, snow height and liquid water content simultaneously. Schmid et al. (2015) is not suitable as reference in l.20 and Heilig et al. (2009) not for SWE measurements. Besides citing Steiner et al. (2018), Henkel et al. (2018, TGRS) and Koch et al. (2019, WRR) should be added as references in l.20. Besides snow accumulation, the GPS techniques derive snow properties under snow ablation/melt conditions. Additionally, in the latter reference, it was possible to derive three snow cover properties (SWE, snow depth and LWC) simultaneously with only one sensor setup.

- In general, a statement on remote sensing approaches to derive snow cover properties in alpine areas is missing (e.g. l. 31ff) – please give a short overview on such techniques.

- p.3, l.23: I would not name it 'in a second application'. This is rather a further type of validation (besides your manual SWE measurements) for CRS SWE.

2. Study site & 3. Data

- I would suggest to merge sections 2 and 3.

- p.4, l.2: Although you have mentioned the altitude of your study site in the introduction, this should definitely also be mentioned in this section.

- p.4, l.10f: How fast does the glacier move? Is there an effect on the measurements (e.g. on the SD sensor installed on a pole)?

4. Methods

- I would suggest including Subsection 4.1 in Section 3.

- The title of Subsection 4.2 might be misleading – it would be better to directly refer to CRS SWE and generally separate between SWE and snow density derivation.

- p.7, l.26: Please insert a reference for the empirical parameters.

- p.8, Table 3: Not sure if it really makes sense and is sound to use for the gap filling different meteorological parameters from different stations (e.g. temperature from station a, humidity from station b etc.). In my opinion, rather one station with an overall best fit of all parameters should be used. Please state on this.

- p.8, l.1: Please use just one unit for SWE (either mm or kg/m$^2$). Regarding SD – in the figures, you use [cm] and here you define SD in [m] – this should be uniform throughout the paper.

- p.9, Fig.2.a: Actually, no red or black crosses are visible in the figure (only red and grey horizontal lines) – please state on this and/or correct. Moreover, the error bars are not really readable. A revised version of this figure would be helpful (it could make sense to display the error bars in a separate figure).

- p.10, l.8: Please introduce N or do you mean Ni?

- p.10, l.10: Why did you chose +/- 1cm? Can this be underlain with a reference?

- p. 10, l.19: In an earlier section you mentioned 4.8 m instead of 4.75 m – please unify.

5. Results

- p.14, Fig.3: Please describe the vertical dashed lines in the figure caption or in a legend. In general, this figure would benefit to be displayed larger (if possible).

- p.15, Fig.4: I really like this figure!

- p.17, l.22: You should underlie the statement of rain gauge undercatch with a reference.

6. Discussion

- Please add the following points in the discussion: Is there a general SWE limit by using CRS? How big is the footprint of the sensor and which shape does it have (e.g.

[Figure]

conical)?

- p.22, l.12: Please specify why there might be problems between 90 and 120 cm.

- p.22, l.26-30: Please insert references in this paragraph.

7. Conclusion

- As this study investigates to a quite big extent the development of the snow density at your study site, this should also be mentioned more prominently in the conclusions section.

Appendix A

- In my opinion the appendix should be integrated in the methods section.

- p.25, l.12: Please introduce N also in the text.

---

## Referee Comment (RC2) · Anonymous Referee #2 · 28 Jun 2019

This paper evaluates the snow accumulation on the Plaine Morte glacier by means of a buried cosmic-ray neutron probe (CRNS) and an approach based on the scaling of the precipitation records of nearby meteorological stations. The accuracy of the field data is assessed by the propagation of possible error sources. Together with the combined approach using different types of field data, this gives important insights into the evolution of the snow pack on the glacier.

The language of the paper is appropiate, as are the figures and tables. Partly, the paper would benefit from considering a geographically broader view on the state-of-the-art as many references focus on Switzerland.

[Figure]

In principle, the paper is suitable for publication in this journal. In particular, the added value of the paper lies in applying a buried CRNS together with other measurements for continuously monitoring the snow accumulation of a mountain glacier. However, prior to further consideration for publication, the following two major concerns need to be addressed carefully:

(1) The story line of the paper needs to be clarified. The title and the final conclusions do not match well with the analysis made. Furthermore, the second part of the analysis is not (yet) connected to the rest of the paper. One could think of some logical links between the two parts, but it is important to state this more clearly, and to frame the rest of the paper accordingly. In addition, it would help the reader if the novelty would be more prounced in the abstract and the conclusions. (2) While the error propagation of the snow depth, snow density and the meteorological measurements is reasonable and covers all important sources of uncertainty, this is not the case for the CRNS data. Most notably, the instrument's precision is most likely largely overestimated. Furthermore, a decrease of the error with increasing SWE is highly unlikely with mostl likely the opposite behaviour being the case. Currently, only the uncertainty of the neutron count rate is considered, and a constant error is added despite the high non-linearity of the signal. The latter is probably the reason why the relative accuracy seems to increase with higher snow accumulation values. The statistical error of neutron count rate itself is an important element of measurement uncertainty, but it refers to uncorrected variations only. The uncorrected count rate includes variations not only of the accumulated SWE but also variations of incoming neutrons, atmospheric pressure, and in amtospheric moisture. An error propagation should thus include the uncertainty of (1) the neutron count uncertainty as already done, (2) the uncertainty of the measurements used for the corrections (Jungfraujoch neutron monitor data, atmospheric pressure, atmospheric moisture), (3) the uncertainty in the parameterisation of the correction functions (e.g., the value for the attenuation lenght, which may vary in space and time), and (4) the uncertainty in the (not well documented) empirical function relating neutron counts to SWE. In total, from figure 2 the error seems to be rather in the

range of 10 to 20 % (and thus around ten times larger than estimated in the paper!), with an increasing trend for high SWE values. Also the comparison with the manual measurements (figure 3) shows that the SWE from CRNS is mostly only toughing the uncertainty bands of the manual measurements, while is partly entirely off. With the current focus of the paper the lack of a proper error propagation of the CRNS data constitutes a severe issue, as the evaluation and the precision of the CRNS are stated prominently in the title and conclusions.

Still, it is interesting to see the application of CRNS for glacier monitoring and I agree with the authors that it constitutes a very promising technique for continuous accumulation meaurements on glaciers. Existing uncertainties should, however, be kept in mind instead of propagating an unrealistically high precision of the SWE estimate.

I believe there are two equally legitimate strategies on how the authors could address this. One is a true and rigerous error propagation with regard to all relevant uncertainty sources of the CRNS SWE estimate. Another could lie in drawing the readers's attention to the fact, that the uncertainty range could be substantially (up to ten times) larger, combined with reframing the paper towards the application rather than the error propagation.

Minor specific remarks:

Page 3 / Line 33-34: Check the sentence ("..define three different scaling factors, one for..."?).

Page 4/ Line 2: It would be helpful when the elevation of the glacier and the sourrounding mountain peaks would be added here.

Page 5/ Line 18: Can you add a few key facts on how the gridded products is produced. Does it contain station data? If so, how reliable is it when the nearby stations have data gaps?

Page 6/ Table 2: Think of readers that are not familar with the Swiss coordinate system.

I'd recommend converting the station coordinates into a globally used system like UTM or WGS84 (lat/lon). In any case, add also the EPSG-code of the coordinate system.

Page 7 / Line 1: The reliability of the CRNS is one of the objectives, thus could cannot be claimed beforehand.

Page 23 / Line 2: the effect is related to SWE not to density.

Page 23 / Line 8: Here, too much confidence is set into CRNS.

---

## Author Comment (AC1) · 22 Aug 2019

**Author Response to Reviews of**

**Evaluating continuous and autonomous snow water equivalent measurements by a cosmic ray sensor on a Swiss glacier**

Rebecca Gugerli, Nadine Salzmann, Matthias Huss and Darin Desilets *The Cryosphere Discussion*, doi:10.5194/tc-2019-106

**RC:** *Reviewer Comment*, AR: *Author Response*,

Manuscript text

**Anonymous Referee #1**

We would like to thank the anonymous referee #1 for his/her time and the thoughtful and constructive review, which significantly improves our manuscript.

Following the suggestions by referee #1 and #2, we decided to adapt the title of the paper to "Continuous and autonomous snow water equivalent measurements by a cosmic ray sensor on an Alpine glacier."

**General comments**

- RC: This paper presents the application of a sub-merged cosmic ray sensor (CRS) on a Swiss glacier to derive daily snow water equivalent (SWE) values for two winter seasons. An additionally installed snow depth (SD) sensor was used to calculate the snow density by CRS SWE and SD. For validation, some manual field measurements were conducted within the two years and precipitation recordings from nearby weather stations as well as a gridded precipitation product were scaled to compare them with the measured CRS SWE. The measurement results derived by CRS are very plausible for snow accumulation, densification and ablation phases. In general, this paper is well written and is, in my opinion, a good contribution to this journal. All measurements are well described and indicated by potential uncertainties and illustrated by significant figures. However, the main focus/ objective of this paper has to be better defined. Are you rather interested in gaining better snow density information or are you mainly focusing on using and validating CRS measurements especially on such a glacial test site, or both? Please emphasize on this – maybe also the title has to be changed accordingly. In some paragraphs, references should be added or revised. Below, I indicated some other moderate to minor issues.
- AR: Thank you for your careful assessment and your constructive feedback. We have investigated the application of the cosmic ray sensor for measurements at challenging sites such as a glacier. The cosmic ray sensor (CRS) has been previously evaluated in the Alpine region (Schattan et al., 2017) and on the Greenland ice sheet (Howat et al., 2018). To complement these studies, we assessed the application of this device on an Alpine glacier. The primary focus on this study lies on the measurement setup and what new knowledge we gain from such observations. To state this more clearly, we re-wrote the end of the introduction.

**Specific comment**

RC: Please use the same units throughout the paper. SWE is usually given in mm (or kg/m2), not in cm w.e.

AR: We changed the units for snow water equivalent (SWE) to mm w.e., but kept the unit for snow depth (SD) as cm.

**Abstract**

- RC: p. 1, l 12-16: The aspect of the comparison of the cosmic ray SWE values and the scaled precipitation is represented quite dominant in the abstract. I think this aspect can be reduced to two 2 sentences and the abstract should better include also a statement on the general applicability.
- AR: We reduced the concerned part in the abstract as follows.

Moreover, we compare daily SWE amounts to precipitation sums from three nearby weather stations located at lower elevations, and to a gridded precipitation dataset. We determine the best possible scaling factor for these precipitation estimates in order to reproduce the measured accumulation on the glacier. Using only one scaling factor for the whole time series, we find a mean absolute error of less than 8 cm w.e. for the reproduced snow accumulation. By applying temperature-specific scaling factors, this mean absolute error can be reduced to less than 6 cm w.e. for all stations. The continuous SWE measurements can be used, for example, to validate scaled precipitation measurements from nearby rain gauges at lower elevations. With these, we can reproduce snow accumulation with a mean absolute error of less than 80 mm w.e.

**1. Introduction**

- RC: In general, a statement on remote sensing approaches to derive snow cover properties in alpine areas is missing (e.g. l. 31ff) please give a short overview on such techniques.
- AR: We included a paragraph on remote sensing approaches.

Besides the in situ manual and autonomous measurements, spaceborn sensors can estimate snow cover, SWE and SD. While these sensors have a large spatial coverage, their resolution is coarse. In addition, estimates of SWE are strongly affected by snow properties such as the snow crystals and the liquid water content which make accurate estimations challenging (Clifford, 2010, Dietz et al., 2012).

- RC: p.2, l.2: Not only the cold and windy conditions are a big challenge for in situ snow measurements in high mountains; please add that they are also often limited by difficult accessibility, complex terrain etc.
- AR: We added a further sentence to explain the limited accessibility and complex terrain.

Cold and windy conditions pose the main challenge for accurate measurements (Sevruk et al., 2009, Rasmussen et al., 2012, Kinar and Pomeroy, 2015). The complex topography and limited accessibility add further challenges in high mountain regions.

RC: p.2, l.13-21: The statements in this paragraph should be revised carefully as some statements are not correct. Some explanations are given in the following: Schmid et al. (2014) combined the upGPR with a

snow depth sensor to additionally derive the liquid water content in snow. The main reasons for combining upGPR travel time with GPS signal strength in Schmid et al. (2015) were to eliminate an overestimation in snow depth during wet snow conditions, which would be the case by using only upGPR measurements, and to be independent of poles as both sensors were buried beneath the snowpack, which could be useful, e.g., in avalanche prone slopes. Moreover, with this upGPR-GPS sensor combination it was possible for the first time to derive SWE, snow height and liquid water content simultaneously. Schmid et al. (2015) is not suitable as reference in 1.20 and Heilig et al. (2009) not for SWE measurements. Besides citing Steiner et al. (2018), Henkel et al. (2018, TGRS) and Koch et al. (2019, WRR) should be added as references in 1.20. Besides snow accumulation, the GPS techniques derive snow properties under snow ablation/melt conditions. Additionally, in the latter reference, it was possible to derive three snow cover properties (SWE, snow depth and LWC) simultaneously with only one sensor setup.

AR: We revised this paragraph carefully, and changed it as follows.

Ground-penetrating radar (GPR) is another method to determine snow accumulation and has been used in various studies (e.g. Heilig et al., 2009, 2010). Schmid et al. (2014) combine a snow depth (SD) sensor with an upward looking GPR (upGPR) installed within the ground below the snowpack. This combination results in continuous estimates of liquid water content, SD and SWE at a high temporal resolution. SWE derived from this method lies within  $\pm 5\%$  discrepancy from manual measurements. In a follow-up study, Schmid et al. (2015) combined an operational upGPR with a low-cost GPS to render the approach independent from additional sensors (e.g. SD). Despite the good agreement with manual measurements of SWE, the underlying algorithm to derive SWE from the upGPR is still prone to errors. For instance, a deviation of 10% in SD may lead to an overor underestimation of 30-40% of the resulting SWE (Schmid et al., 2014). Furthermore, erroneous identifications of the reflection horizons affect the resulting SWE (Heilig et al., 2009; Schmid et al., 2015).

Ground-penetrating radar (GPR) systems, which are installed below the snowpack and look upwards, provide information about the snow stratigraphy (Heilig et al., 2009) and SD (Heilig et al., 2010, Schmid et al., 2014). Combined with a low-cost GPR, Schmid et al. (2015) derived the liquid water content, SD and SWE without additional information and independent of installation poles which makes it suitable for avalanche prone slopes. In more recent studies, Steiner et al. (2018) and Henkel et al. (2018) present sub-snow low-cost GPS as a promising method to continuously derive SWE. Koch et al. (2019) extend these studies by using the GPS signals to additionally derive SD and liquid water content. However, the processing of the GPS signal is relatively complex and different for dry- and wet-snow (Koch et al., 2019).

**RC: p.3, l.23: I would not name it in a second application. This is rather a further type of validation (besides your manual SWE measurements) for CRS SWE.**

AR: In general, we have more confidence in the CRS SWE than in the scaled precipitation measurements. For this reason, we scaled the precipitation events to fit the accumulated snow on the glacier and validate it with the CRS SWE measurements.

**2. Study Site & 3. Data**

- **RC:** I would suggest to merge sections 2 and 3.
- AR: We added the section "Study site" under 3. Data.
- **RC:** p.4, l.2: Although you have mentioned the altitude of your study site in the introduction, this should definitely also be mentioned in this section.
- AR: We agree and added the altitude of the installation location. In addition, we added the elevation bands of the glacier in the text as follows.

Plaine Morte is particular in that it has almost no elevation gradient. In fact, most of the glacier surface is located between 2650 masl and 2800 masl (GLAMOS, 2017). With a surface area of 7.4 km2 it is the largest plateau glacier in the European Alps.

- RC: p.4, l.10f: How fast does the glacier move? Is there an effect on the measurements (e.g. on the SD sensor installed on a pole)?
- AR: Due to the flatness of the glacier, the surface velocity is small (in the order of 2-5 m per year, Huss et al., 2013). Especially at the site of installation, glacier movement is negligible and does not affect the measurements.

**4. Methods**

- **RC:** I would suggest including Subsection 4.1 in Section 3.
- AR: We understand this point, but we decided against including subsection 4.1 into Section 3 because we consider it a method rather than data.
- **RC:** The title of Subsection 4.2 might be misleading it would be better to directly refer to CRS SWE and generally separate between SWE and snow density derivation.
- AR: We split this section into two sections. The first one is about calculating SWE from neutron counts and its uncertainties, and the second one is about calculating the bulk snow density and its uncertainties.
- **RC:** *p.7, l.26: Please insert a reference for the empirical parameters.*
- AR: The empirical function and its paramteres have been previously used by Howat et al. (2018). We added this reference accordingly.
- **RC:** p.8, Table 3: Not sure if it really makes sense and is sound to use for the gap filling different meteorological parameters from different stations (e.g. temperature from station a, humidity from station b etc.). In my opinion, rather one station with an overall best fit of all parameters should be used. Please state on this.
- AR: We absolutely understand this point of view. With regard to physical consistency, it makes sense to only use

one station. Because we have the independent SWE measurements, the physical consistency of the gap fill is not that important here. Snow depth is the most important parameter with missing data. Thus, we wanted the best-possible correlation for snow depth (SLFGA2). The correlation for temperature and relative humidity does not differ greatly between SLFGA2 and SLFDIA. But it differs for wind speeds. Therefore, we decided to use temperature and humidity from SLFDIA, while using wind speeds from SLFGU2. In general, wind components are more variable in complex terrain than temperature and relative humidity. Given the good correlation of SD with SLFGA2, the effects of wind speeds on the snow dynamics do not seem that important. In conclusion, we assume that the best choice for this study is to use different stations for the gap filling. Concerning the results, the results in Fig.7 a, b would remain similar because the correlation is in the similar range. For Fig.7d, it does not change because the gap is not presented due to the missing wind direction (where we decided not to fill the gap due to the high variability).

**RC: p.8, l.1: Please use just one unit for SWE (either mm or kg/m2). Regarding SD – in the figures, you use [cm] and here you define SD in [m] – this should be uniform throughout the paper.**

- AR: We adapted the units to mm w.e. for SWE and cm for SD. To make the equation consistent with the given units, we added a conversion constant (c) with a value 100 cm  $m^{-1}$  to Eq.3.
- RC: p.9, Fig.2.a: Actually, no red or black crosses are visible in the figure (only red and grey horizontal lines) – please state on this and/or correct. Moreover, the error bars are not really readable. A revised version of this figure would be helpful (it could make sense to display the error bars in a separate figure).
- AR: The crosses are not visible because the scale is too large for the uncertainties which are given by the raw neutron count. We replaced Fig.2 with the following figure. The uncertainties are discussed separately.

Figure 1: Grey dots are the uncorrected hourly neutron counts and black dots the uncorrected daily means. The orange dots represent the corrected daily means. Blue dots show SWE from the field data and the corresponding neutron counts of the field work days.

**RC: p.10, l.8: Please introduce N or do you mean Ni?**

AR: N refers to  $N_i$ . We will correct it in Eq. 6. and Eq. A.1

**RC: p.10, l.10: Why did you chose +/- 1cm? Can this be underlain with a reference?**

AR: This +/- 1 cm originates from an analysis during snow free conditions (not presented here), so it cannot be underlain with a reference. Nevertheless, we changed the uncertainty estimate of the CRS SWE as suggested by referee #2 to make it more sound. In the new approach, no systematic bias is added.

**RC: p. 10, l.19: In an earlier section you mentioned 4.8 m instead of 4.75 m – please unify.**

AR: We changed it to 4.8 m consistently.

**5. Results**

RC: p.14, Fig.3: Please describe the vertical dashed lines in the figure caption or in a legend. In general, this figure would benefit to be displayed larger (if possible).

- AR: We added a sentence describing the dashed and dash-dotted line. We also revised this figure to improve the layout.
- RC: p.15, Fig.4: I really like this figure!
- AR: *Thank you*.
- RC: p.17, l.22: You should underlie the statement of rain gauge undercatch with a reference.
- AR: We added a reference of a study that aims at quantifying the undercatch of rain gauges as proposed in the following.

As to be expected, precipitation sums are significantly lower than snow accumulation on the glacier. This is mainly due to orographic effects but could also be caused by snow drift, or undercatch of the rain gauge (e.g. Pollock et al., 2018).

**6. Discussion**

- **RC:** Please add the following points in the discussion: Is there a general SWE limit by using CRS? How big is the footprint of the sensor and which shape does it have (e.g. conical)?
- AR: We added these points in the discussion as follows.

The SWE limit is given by the exponential nature of the conversion equation (neutron counts to SWE). The uncertainty increases with lower neutron counts and consequently a lower  $N_i$  to  $N_0$  ratio. At some point, the relative neutron count is no longer sensitive to SWE, and that corresponds to the limits of the device. Additionally, with decreasing neutron counts the noise level becomes large enough to loose the signal.

The footprint of the CRS measurements are difficult to define because it is still not well understand how the neutrons are dispersed within the snowpack. Schattan et al. (2017) give an estimate of around 230 to 270 m, but their footprint is based on an above ground CRS. As the instrument in this study lies below the snow pack, the footprint is different. The CRS measures from all sides and with the dispersive nature of the neutrons, measures the surrounding snow pack. Most likely, the footprint is in the range of a couple of meters around the CRS, but further studies are needed to investigate this.

**RC: p.22, l.12: Please specify why there might be problems between 90 and 120 cm.**

AR: Two field campaigns were conducted where SWE amounts were between 90 and 120 cm. Both field measurements differ by more than 15% from the CRS SWE. One of the field measurements is largely overestimated while the other one is underestimated. Because the spread of these two measurements is so large around the fitted curve, we assume that there might be a systematic bias in one of the corrections during one of the field campaigns rather than a misfit. We added this point in the discussion.

**RC:** *p.22, l.26-30: Please insert references in this paragraph.**

AR: We added some references accordingly.

In general, one could argue that a snow scale or snow pillow would have been more suitable for such an analysis, especially given the higher temporal resolution during the accumulation phase. However, they require a large flat surface (e.g. Egli et al., 2009, Kinar and Pomeroy, 2015). This is not given at our study site because of the surface roughness of the ice, the lack of a large flat surface and the changing surface by ice melt during the summer season. In addition, ice bridging would have been problematic. For sub-snow GPS, potential problems might occur due to glacier ice flow or the surface melting which would affect the position of the GPS. In addition, data processing is different for dry and wet snow and is therefore more complicated (Koch et al., 2019).

**7. Conclusion**

- **RC:** As this study investigates to a quite big extent the development of the snow density at your study site, this should also be mentioned more prominently in the conclusions section.
- AR: We added the following paragraph about density evolution in the conclusions.

With this measurement setup, we could show that all phases of the snow pack from accumulation over densification to ablation are distinguishable because of the pronounced relation between SD and SWE. Throughout the accumulation phase, snow densities are low with periodical cycles of snowfall and subsequent densification. Densities reach maximum values around 400 kg m-3. When the maximum in SWE is reached, the snow pack begins to densify continuously before it begins to melt. During the ablation phase, densities remain almost constant.

**Appendix A**

- **RC:** In my opinion the appendix should be integrated in the methods section.
- AR: We agree and integrated it in Section 4.2.

**RC:** *p.25, l.12: Please introduce N also in the text.**

AR: We will replace N with  $N_i$  to render the equations consistent.

**References**

- Clifford, D. (2010). Global estimates of snow water equivalent from passive microwave instruments: History, challenges and future developments. *International Journal of Remote Sensing*, 31(14):3707–3726.
- Dietz, A. J., Kuenzer, C., Gessner, U., and Dech, S. (2012). Remote sensing of snow a review of available methods. *International Journal of Remote Sensing*, 33(13):4094–4134.
- Egli, L., Jonas, T., and Meister, R. (2009). Comparison of different automatic methods for estimating snow water equivalent. *Cold Regions Science and Technology*, 57(2-3):107–115.

- GLAMOS (2017). The swiss glaciers 2013/14-2014/15, glaciological reports no 135-136, yearbooks of the cryospheric commission of the swiss academy of sciences (scnat), published since 1964 by vaw/ eth zurich.
- Heilig, A., Eisen, O., and Schneebeli, M. (2010). Temporal observations of a seasonal snowpack using upward-looking gpr. *Hydrological Processes*, 24(22):3133–3145.
- Heilig, A., Schneebeli, M., and Eisen, O. (2009). Upward-looking ground-penetrating radar for monitoring snowpack stratigraphy. *Cold Regions Science and Technology*, 59(2-3):152–162.
- Henkel, P., Koch, F., Appel, F., Bach, H., and Prasch, M. (2018). Snow Water Equivalent of Dry Snow Derived From GNSS Carrier Phases. *IEEE transactions on geoscience and remote sensing*, 56(6):3561–3572.
- Howat, I. M., De La Peña, S., Desilets, D., and Womack, G. (2018). Autonomous ice sheet surface mass balance measurements from cosmic rays. *Cryosphere*, 12(6):2099–2108.
- Kinar, N. J. and Pomeroy, J. W. (2015). Measurement of the physical properties of the snowpack. *Reviews of Geophysics*, 53(2):481–544.
- Koch, F., Henkel, P., Appel, F., Schmid, L., Bach, H., Lamm, M., Prasch, M., Schweizer, J., and Mauser, W. (2019). Retrieval of Snow Water Equivalent, Liquid Water Content, and Snow Height of Dry and Wet Snow by Combining GPS Signal Attenuation and Time Delay Water Resources Research. *Water Resources Research*, pages 4465–4487.
- Pollock, M. D., O'Donnell, G., Quinn, P., Dutton, M., Black, A., Wilkinson, M. E., Colli, M., Stagnaro, M., Lanza, L. G., Lewis, E., Kilsby, C. G., and O'Connell, P. E. (2018). Quantifying and mitigating wind-induced undercatch in rainfall measurements. *Water Resources Research*, 54(6):3863–3875.
- Rasmussen, R., Baker, B., Kochendorfer, J., Meyers, T., Landolt, S., Fischer, A. P., Black, J., Thériault, J. M., Kucera, P., Gochis, D., Smith, C., Nitu, R., Hall, M., Ikeda, K., and Gutmann, E. (2012). How well are we measuring snow: The NOAA/FAA/NCAR winter precipitation test bed. *Bulletin of the American Meteorological Society*, 93(6):811–829.
- Schattan, P., Baroni, G., Oswald, S. E., Schöber, J., Fey, C., Kormann, C., Huttenlau, M., and Achleitner, S. (2017). Continuous monitoring of snowpack dynamics in alpine terrain by aboveground neutron sensing. *Water Resources Research*, 53(5):3615–3634.
- Schmid, L., Heilig, A., Mitterer, C., Schweizer, J., Maurer, H., Okorn, R., and Eisen, O. (2014). Continuous snowpack monitoring using upward-looking ground-penetrating radar technology. *Journal of Glaciology*, 60(221):509–525.
- Schmid, L., Koch, F., Heilig, A., Prasch, M., Eisen, O., Mauser, W., and Schweizer, J. (2015). A novel sensor combination (upGPR-GPS) to continuously and nondestructively derive snow cover properties. *Geophysical Research Letters*, pages 3397–3405.
- Sevruk, B., Ondrás, M., and Chvíla, B. (2009). The WMO precipitation measurement intercomparisons. *Atmospheric Research*, 92(3):376–380.
- Steiner, L., Meindl, M., Fierz, C., and Geiger, A. (2018). An assessment of sub-snow GPS for quantification of snow water equivalent. *The Cryosphere*, 12(10):3161–3175.

---

## Author Comment (AC2) · 22 Aug 2019

**Author Response to Reviews of**

**Evaluating continuous and autonomous snow water equivalent measurements by a cosmic ray sensor on a Swiss glacier**

Rebecca Gugerli, Nadine Salzmann, Matthias Huss and Darin Desilets *The Cryosphere Discussion*, doi:10.5194/tc-2019-106

**RC:** *Reviewer Comment*, AR: *Author Response*,

Manuscript text

**Anonymous Referee #2**

We would like to thank the anonymous referee #2 for his/her time and the thoughtful and constructive review, which significantly improves our manuscript.

Following the suggestions by referee #1 and #2, we decided to change the title of the paper to "Continuous and autonomous snow water equivalent measurements by a cosmic ray sensor on an Alpine glacier".

**General comments**

- RC: This paper evaluates the snow accumulation on the Plaine Morte glacier by means of a buried cosmicray neutron probe (CRNS) and an approach based on the scaling of the precipitation records of nearby meteorological stations. The accuracy of the field data is assessed by the propagation of possible error sources. Together with the combined approach using different types of field data, this gives important insights into the evolution of the snow pack on the glacier. The language of the paper is appropriate, as are the figures and tables. Partly, the paper would benefit from considering a geographically broader view on the state-of-the-art as many references focus on Switzerland. In principle, the paper is suitable for publication in this journal. In particular, the added value of the paper lies in applying a buried CRNS together with other measurements for continuously monitoring the snow accumulation of a mountain glacier.
- AR: Thank you for your valuable assessment and the interesting feedback. The state-of-the-art has many, but not only Swiss references. But, we admit that there has been a strong focus on the Swiss observational network. This is to make a direct comparability within the same region easier. Nevertheless, we broadened the state-of-art section by adding more non-Swiss references.

**RC:** However, prior to further consideration for publication, the following two major concerns need to be addressed carefully:**

- AR: To address these two major concern in more detail, we split the following comment of referee #2 into smaller sections. That allows us to directly address each comment point by point.
- RC: (1) The story line of the paper needs to be clarified. The title and the final conclusions do not match well with the analysis made. Furthermore, the second part of the analysis is not (yet) connected to the rest of the paper. One could think of some logical links between the two parts, but it is important to state this

**more clearly, and to frame the rest of the paper accordingly. In addition, it would help the reader if the novelty would be more pronounced in the abstract and the conclusions.**

AR: The main focus of the paper is to assess the application of a CRS in combination with the sonic ranging sensor for continuous snow water equivalent (SWE) and snow depth (SD) measurements on glaciers, and to show the advantages of such a measurement setup. We discuss the advantages of such a measurement setup and show what kind of knowledge we can gain, also in regard to precipitation scaling at high altitudes. Applying a scaling factor on precipitation estimates is a common approach when no in situ measurements are available. We re-framed the story line of the paper accordingly, following this story line. In addition, we rephrased the end of the introduction explaining more clearly the links between the CRS measurements and the precipitation scaling.

**RC: (2) While the error propagation of the snow depth, snow density and the meteorological measurements is reasonable and covers all important sources of uncertainty, this is not the case for the CRNS data. Most notably, the instrument's precision is most likely largely overestimated.**

- AR: We would like to point out that we differ between precision and accuracy, which we both call an uncertainty. With the error propagation, we calculate the precision of the instrument. With the field data, we could validate the CRS measurements and show that it slightly overestimate SWE amounts in average within  $\pm 13\%$ . So our accuracy is rather  $\pm 13\%$  than what the precision analysis showed. In addition, we believe that the comparison to the independent field data is more valuable than the propagated precision of the CRS. We will emphasize this point more strongly in the paper. In addition, we calculated an error propagation taking into account all relevant sources of uncertainty. More information is provided with the following comments.
- RC: Furthermore, a decrease of the error with increasing SWE is highly unlikely with mostly likely the opposite behaviour being the case. Currently, only the uncertainty of the neutron count rate is considered, and a constant error is added despite the high non-linearity of the signal. The latter is probably the reason why the relative accuracy seems to increase with higher snow accumulation values.
- AR: Fig. 2 shows that the absolute uncertainty  $\sigma_{crs}$  increases with higher SWE amounts. However, the rate of increase changes which affect the relative uncertainty. We also point out that in the discussion we wrote "Generally higher neutron counts and lower SWE values place Howat et al. (2018) on a steeper part of the calibration curve which results in more precise results." Howat et al. (2018) use the same measurement device with the same empirical function and parameters. Since we redid the error propagation and removed the constant factor, we also replaced Fig.2 accordingly.
- **RC:** The statistical error of neutron count rate itself is an important element of measurement uncertainty, but it refers to uncorrected variations only. The uncorrected count rate includes variations not only of the accumulated SWE but also variations of incoming neutrons, atmospheric pressure, and in atmospheric moisture.
- AR: In the new error propagation, we consider the effect of incoming neutrons and pressure. However, we expect the effect of atmospheric moisture on the count rate to be minimal (once the probe is buried). Work with the CRNS as a soil moisture sensor suggests that atmospheric moisture may affect  $N_0$  by some small amounts. But this is actually not known (Rosolem et al., 2013).
- **RC:** An error propagation should thus include the uncertainty of (1) the neutron count uncertainty as already done, (2) the uncertainty of the measurements used for the corrections (Jungfraujoch neutron monitor

data, atmospheric pressure, atmospheric moisture), (3) the uncertainty in the parameterisation of the correction functions (e.g., the value for the attenuation length, which may vary in space and time), and (4) the uncertainty in the (not well documented) empirical function relating neutron counts to SWE. In total, from figure 2 the error seems to be rather in the range of 10 to 20% (and thus around ten times larger than estimated in the paper!), with an increasing trend for high SWE values. Also the comparison with the manual measurements (figure 3) shows that the SWE from CRNS is mostly only touching the uncertainty bands of the manual measurements, while is partly entirely off.

AR: We added the following part to the manuscript taking into account not only the uncertainty of the neutron count itself, but also of the measurements used to correct that neutron count. We admit that the precision is not equal to the accuracy of the CRS, and that claiming such an high accuracy as we have a precision would be too optimistic. However, we believe that a proper uncertainty estimate of the empirical function and its parameterisation would provide enough material for a paper on its own. Therefore, we consider it to be too much for the scope of this paper. But, we added a section in the discussion, where we critically discuss all sources of uncertainties, mainly points (3) and (4) given by referee #2. With this approach, we hope to address this major concern appropriately.

The calculated SWE amounts depend only on the relative neutron counts as the empirical equation is fitted for relative neutron counts. Therefore, we base our error propagation on all corrections applied to the raw neutron count up to the calculation of the relative neutron count. For the corrections we use

$$N_{\rm rel,i} = \frac{N_i}{N_0} = N_{\rm raw,i} \cdot \left(\beta \cdot \left(\frac{F_{\rm inc,i}}{F_{\rm inc,0}} - 1\right) + 1\right) \cdot exp\left(\frac{p_i - p_0}{L}\right) \cdot \frac{1}{N_0} \tag{1}$$

The raw neutron counts  $N_{\text{raw}}$ , the incoming neutron flux ( $F_{\text{inc},i}$ ) and air pressure ( $p_i$ ) change with time, but remain independent from each other. Following the rules of error propagation of a non-linear equation, we approximate the uncertainty of  $N_{\text{rel},i}$  as

$$\begin{split} \sigma_{N_{\rm rel,i}}^2 &\approx \left(\frac{\partial N_{\rm rel,i}}{\partial N_{\rm raw,i}}\right)^2 \cdot \sigma_{N_{\rm raw,i}}^2 + \left(\frac{\partial N_{\rm rel,i}}{\partial N_0}\right)^2 \cdot \sigma_{N_0}^2 \\ &+ \left(\frac{\partial N_{\rm rel,i}}{\partial F_{\rm inc,i}}\right)^2 \cdot \sigma_{F_{\rm inc,i}}^2 + \left(\frac{\partial N_{\rm rel,i}}{\partial F_{\rm inc,0}}\right)^2 \cdot \sigma_{F_{\rm inc,0}}^2 + \left(\frac{\partial N_{\rm rel,i}}{\partial \beta}\right)^2 \cdot \sigma_{\beta}^2 \\ &+ \left(\frac{\partial N_{\rm rel,i}}{\partial p_i}\right)^2 \cdot \sigma_{p_i}^2 + \left(\frac{\partial N_{\rm rel,i}}{\partial p_0}\right)^2 \cdot \sigma_{p_0}^2 + \left(\frac{\partial N_{\rm rel,i}}{\partial L}\right)^2 \cdot \sigma_{L}^2 \quad (2) \end{split}$$

The uncertainty  $\sigma_{N_{\text{rel},i}}^2$  is then propagated through

$$SWE_i = -\frac{1}{\Lambda} \cdot \ln N_{\rm rel,i} \tag{3}$$

to estimate the uncertainty  $\sigma_{\rm crs}$

$$\sigma_{\rm crs} \approx sqrt \left(\frac{\partial SWE_i}{\partial N_{rel,i}}\right)^2 \cdot \sigma_{N_{rel,i}}^2 \tag{4}$$

For each variable, we attributed an uncertainty to the best of our knowledge and supported by literature. The uncertainties are not always known, therefore we assume rather generous estimates for the uncertainties. Where possible, we base our estimates on literature values. Table 1 gives an overview of these values.

For all neutron counts ( $N_{raw,i}$ ,  $N_0$ ,  $F_{inc,0}$ ,  $F_{inc,i}$ ), we assume the square root for the uncertainties ( $\sigma$ ) (e.g. Zreda et al., 2012, Schrön et al., 2018). With the integration over a time period t, the uncertainty is reduced by  $t^{-0.5}$  (Schrön et al., 2018). While the relative uncertainty of  $N_{raw}$  varies between 1.5%-5.3% for hourly observations, it varies between 0.3%-1% for the integrated daily estimates.

The incoming radiation has a low uncertainty because its precision is with around 190 counts per second very high. However, incoming radiation needs to be corrected for different sites. This is done with  $\beta$ . Our study site is located less than 40 km (air line) from the Jungfraujoch and is around 900 m lower. Hence, this correction factor ( $\beta$ ) is rather small for our site (0.95) and so is its uncertainty ( $\sigma_{\beta}$ , 0.03).

The uncertainty given by air pressure  $(\sigma_{p_i}, \sigma_{p_0})$  is based on the instruments precision which is 0.1 hPa (Lufft, 2019). For the mass attenuation length L, we use 132 hPa with an uncertainty of  $\pm 2$  hPa ( $\sigma_L$ ). This uncertainty corresponds to the difference of shielding depths from latitudes north and south of Switzerland as shown in Fig.1 of Andreasen et al. (2017).

To render the error propagation more robust, we calculated  $\sigma_{crs}$  using three setups with each the observations and with a dataset where the time dependent variables  $(N_{raw,i}, p_i, F_{inc,i})$  are uniformly sampled within boundaries defined over their minma and maxima values. This semi-artificial dataset encompasses  $4.8 \cdot 10^8$  data points.

In the first setup (Fig. 1a,d), the uncertainty is calculated for an hourly resolution. In a second setup (Fig. 1b,e), we consider the integrated daily observations with their given uncertainties (Table 1). In Fig. 1c,f, we limited the uncertainties of the neutron counts ( $\sigma_{Finc,i}$ ,  $\sigma_{Nraw,i}$ ) to a minimal uncertainty of 0.5%. Fig. 1d,e,f show the relative contribution of each parameter to the propagated uncertainty. The contribution is quantified as the relative contribution of the uncertainty terms to the overall uncertainty. The uncertainty terms are defined as the squared derivative multiplied by the squared uncertainty (see also Eq. 2). Fig. 1 shows that the main uncertainty is given by the neutron count uncertainty. The almost equal contribution of several parameters in Fig. 1f is a result of the limited minimal uncertainty. Nevertheless, the neutron count uncertainty clearly dominates also in this setup for higher SWE amounts.

| Variables       | hourly observations                    | $\sigma$ (hourly)      | daily observations                     | $\sigma$ (daily)       |
|-----------------|----------------------------------------|------------------------|----------------------------------------|------------------------|
| $N_{\rm raw,i}$ | [354; 4450] cph                        | $\sqrt{N_{\rm raw,i}}$ | $[8.5 \cdot 10^3; 1.1 \cdot 10^5]$ cpd | $\sqrt{N_{\rm raw,i}}$ |
| $N_0$           | 4143 cph                               | $\sqrt{N_0}$           | $9 \cdot 10^4 \text{ cpd}$             | $\sqrt{N_0}$           |
| $F_{\rm inc,i}$ | $[6.6 \cdot 10^5; 7.0 \cdot 10^5]$ cph | $\sqrt{F_{\rm inc,i}}$ | $[1.6 \cdot 10^7; 1.7^7]$ cpd          | $\sqrt{F_{\rm inc,i}}$ |
| $F_{\rm inc,0}$ | $6.9 \cdot 10^5$ cph                   | $\sqrt{F_{\rm inc,0}}$ | $1.6 \cdot 10^7 \text{ cpd}$           | $\sqrt{F_{\rm inc,0}}$ |
| $\beta$         | 0.95                                   | 0.03                   | 0.95                                   | 0.03                   |
| $p_i$           | [708; 747] hPa                         | 0.1 hPa                | [708; 747] hPa                         | 0.1 hPa                |
| $p_0$           | 739 hPa                                | 0.1 hPa                | 739 hPa                                | 0.1 hPa                |
| L               | 132 hPa                                | 2 hPa                  | 132 hPa                                | 2 hPa                  |

Table 1: Table with all given values and uncertainties. The units cph and cpd stand for counts per hour and day, respectively. Brackets show the minimum and maximum within the time series.

Figure 1: (a)-(c) show the absolute uncertainties as a function of SWE amounts. The grey dots represent a sample data and the black dots the observations. (e)-(f) show the relative contribution of each parameter to the overall uncertainty. (a) and (d) present the results based on an hourly temporal resolution, (b) and (e) show the results of integrated daily values and (c) and (f) integrated daily values with a limited minimal uncertainty of 0.5 %.

RC: With the current focus of the paper the lack of a proper error propagation of the CRNS data constitutes a severe issue, as the evaluation and the precision of the CRNS are stated prominently in the title and conclusions. Still, it is interesting to see the application of CRNS for glacier monitoring and I agree with the authors that it constitutes a very promising technique for continuous accumulation measurements on glaciers. Existing uncertainties should, however, be kept in mind instead of propagating an unrealistically high precision of the SWE estimate. I believe there are two equally legitimate strategies on how the authors could address this. One is a true and rigorous error propagation with regard to all relevant uncertainty sources of the CRNS SWE estimate. Another could lie in drawing the reader's attention to the

**fact, that the uncertainty range could be substantially (up to ten times) larger, combined with reframing the paper towards the application rather than the error propagation.**

AR: We integrated a more robust error propagation (see above). We also discussed the accuracy of the CRS with regard to the field data more critically in the coresponding section. Furthermore, we changed the title of the paper to emphasize the focus more on the application.

**Specific comment**

- RC: Page 3/ Line 33-34: Check the sentence ("..define three different scaling factors, one for..."?).
- AR: We adapted the sentence as follows:

In a more elaborate approach, we define three different scaling factor for each weather stations. In a more elaborate approach, we define three different scaling factors, one for liquid precipitation only, one for solid precipitation only and one for mixed-phase precipitation. These scaling factors are defined for each weather station and grid cell individually and depend on in situ hourly temperature measurements.

**RC:** Page 4/Line 2: It would be helpful when the elevation of the glacier and the surrounding mountain peaks would be added here.**

AR: We added the following sentence in the section "Study site".

Our study site is located on the Glacier de la Plaine Morte (in the following: Plaine Morte), where we deployed a CRS along with an automatic weather station (46° 22.8'N, 7°29.7'E, 2690 masl). This glacier is situated on the ridge between two Alpine regions of Switzerland, the Bernese Alps in the North and the Rhône valley in the South (Huss et al., 2013). The glacier is surrounded by mountain peaks from 2926 masl (Pointe de la Plaine Morte) up to 3244 masl (Wildstrubel, see Fig.1)

**RC: Page 5/ Line 18: Can you add a few key facts on how the gridded products is produced. Does it contain station data? If so, how reliable is it when the nearby stations have data gaps?**

AR: We added the following paragraph to address the comment and the question.

RhiresD is based on data from around 400 automatic as well as manual non-realtime quality checked precipitation measurements. In a first step, the climatological mean precipitation measurements for the calendar month of a given day are spatially interpolated. Thereby, regionally varying precipitation - topography relationships are applied. With this interpolation, relative anomalies for a station are calculated for the given day. Adopting a weighting scheme, the relative anomalies are spatially interpolated prior to multiplication with the climatologocial mean field. Main sources of uncertainty are given by the interpolation, rain-gauge measurements, grid spacing and its effective resolution and the temporal variation of the number of stations. For further information, the reader is referred to the technical document provided by MeteoSwiss (MeteoSwiss, 2013).

RC: Page 6/ Table 2: Think of readers that are not familiar with the Swiss coordinate system. I would recommend converting the station coordinates into a globally used system like UTM or WGS84 (lat/lon). In any

**case, add also the EPSG-code of the coordinate system.**

- AR: We added the WGS84 coordinates, but also kept the Swiss coordinate because it is easier to identify the location in the map of Fig.1. Additionally, we added a cross in the lower left corner of Fig.1 with the corresponding WGS84 latitude and longitude coordinates.
- RC: Page 7 Line 1: The reliability of the CRNS is one of the objectives, thus could cannot be claimed beforehand.
- AR: With "reliable" we meant "technically reliable". We adapted the sentence as follows:

Once deployed, the CRS measured *reliably*continuously over the two winter seasons with one exception.

**RC: Page 23/ Line 2: the effect is related to SWE not to density.**

AR: We revised this paragraph carefully, also in regard to all the additional points we added (see above).

**RC: Page 23/ Line 8: Here, too much confidence is set into CRNS.**

AR: In the discussion, we point out all potential influences that could lead to overestimated SWE observations. However, it is also possible that there was a problem with the SD measurements. We tried to give a neutral exposition on both possibilities without pointing towards one or the other instrument. But, it still seems to have been slightly biased towards one direction and therefore, we changed part of that sentence as follows:

[..] Therefore, not all these effects would be identifiable, and explanations remain speculative. Despite all potential explanation for errors by the CRS, it could also be a problem with the SD measurements rather than the SWE measurements. In our study setup, several reasons could cause erroneous SD measurements. Given our study setup, erroneous SD measurements cannot be excluded either.

To underline this statement, we show a photo from the mast taken in June 2019 below. We are not intending to add the photo to the paper manuscript. Between the last field work in April 2019 and this one, the site was not visited, and we encountered it with the large depression around the mast itself.

Figure 2: Photo of the mast installation taken in June 2019.

**References**

- Andreasen, M., Jensen, K. H., Desilets, D., Franz, T. E., Zreda, M., Bogena, H. R., and Looms, M. C. (2017). Status and Perspectives on the Cosmic-Ray Neutron Method for Soil Moisture Estimation and Other Environmental Science Applications. *Vadose Zone Journal*, 16(8):0.
- Howat, I. M., De La Peña, S., Desilets, D., and Womack, G. (2018). Autonomous ice sheet surface mass balance measurements from cosmic rays. *Cryosphere*, 12(6):2099–2108.
- Huss, M., Voinesco, A., and Hoelzle, M. (2013). Implications of climate change on Glacier de la Plaine Morte, Switzerland. *Geographica Helvetica*, 68(4):227–237.
- Lufft (2019). Technical data. ventus-umb ultrasonic wind sensor. Technical report, Mess- und Regeltechnik GmbH.
- MeteoSwiss (2013). Documentation of MeteoSwiss Grid-Data Products Daily Precipitation (final analysis): RhiresD. Technical Report August.
- Rosolem, R., Shuttleworth, W. J., Zreda, M., Franz, T. E., Zeng, X., and Kurc, S. A. (2013). The effect of atmospheric water vapor on neutron count in the cosmic-ray soil moisture observing system. *Journal of Hydrometeorology*, 14(5):1659–1671.
- Schrön, M., Zacharias, S., Womack, G., Köhli, M., Desilets, D., Oswald, S. E., Bumberger, J., Mollenhauer, H., Kögler, S., Remmler, P., Kasner, M., Denk, A., and Dietrich, P. (2018). Intercomparison of cosmic-ray neutron sensors and water balance monitoring in an urban environment. *Geoscientific Instrumentation*, *Methods and Data Systems*, 7(1):83–99.
- Zreda, M., Shuttleworth, W. J., Zeng, X., Zweck, C., Desilets, D., Franz, T., and Rosolem, R. (2012). COSMOS: The cosmic-ray soil moisture observing system. *Hydrology and Earth System Sciences*, 16(11):4079–4099.

---

## Author Response (AR1)

Dear editor,

First, we would like to thank the two referees and the editor for dedicating their time to our manuscript and providing us with an in-depth feedback.

Our major revisions include the following points:

- We changed the title to "Continuous and autonomous snow water equivalent measurements by a cosmic ray sensor on an Alpine glacier"
- We improved (rewrote and rearranged several parts) the introduction by adding the points raised by referee #1, and by adjusting the story line.
- We improved the description of how we process the neutron count rate in Section 3.2 and additionally added a Section 3.4. This section includes an error propagation of the snow water equivalent derived by the cosmic ray sensor following the suggestions by referee #2. We also adapted Fig.2 accordingly and added an additional figure.
- We improved Fig.4 by changing its design (Fig.3 in first submission).
- Following the new estimation of the CRS precision, we adpated our evaluation of the daily changes in SWE and snow depth even though it did not change significantly (Table 6, Fig.7-8, in discussion part (Fig.6-7 and Table 4)
- We rewrote the discussion sections and split them into three sections:
  - Section 5.1 was re-structured and includes all the points raised by referee #1 and #2.
  - Section 5.3 was restructured in view of improving the story line of the paper.
- The conclusion was adapted to the points raised by referee #1 and #2.

Please find a point-by-point answer for each referee attached to this letter followed by all changes made in the original manuscript.

We hope to have addressed all major, moderate and minor concerns.

We thank you for considering the revised manuscript for publication, and look forward to hearing from you.

Rebecca Gugerli, on behalf of the authors

**Author Response to Reviews of**

**Evaluating continuous and autonomous snow water equivalent measurements by a cosmic ray sensor on a Swiss glacier**

Rebecca Gugerli, Nadine Salzmann, Matthias Huss and Darin Desilets
*The Cryosphere Discussion,* `doi:10.5194/tc-2019-106`
* * *
**RC:** *Reviewer Comment*,    AR: *Author Response*,    ☐ Manuscript text

**Anonymous Referee #1**

We would like to thank the anonymous referee #1 for his/her time and the thoughtful and constructive review, which significantly improves our manuscript.

Following the suggestions by referee #1 and #2, we decided to adapt the title of the paper to "Continuous and autonomous snow water equivalent measurements by a cosmic ray sensor on an Alpine glacier."

**General comments**

**RC:** *This paper presents the application of a sub-merged cosmic ray sensor (CRS) on a Swiss glacier to derive daily snow water equivalent (SWE) values for two winter seasons. An additionally installed snow depth (SD) sensor was used to calculate the snow density by CRS SWE and SD. For validation, some manual field measurements were conducted within the two years and precipitation recordings from nearby weather stations as well as a gridded precipitation product were scaled to compare them with the measured CRS SWE. The measurement results derived by CRS are very plausible for snow accumulation, densification and ablation phases. In general, this paper is well written and is, in my opinion, a good contribution to this journal. All measurements are well described and indicated by potential uncertainties and illustrated by significant figures. However, the main focus/ objective of this paper has to be better defined. Are you rather interested in gaining better snow density information or are you mainly focusing on using and validating CRS measurements especially on such a glacial test site, or both? Please emphasize on this – maybe also the title has to be changed accordingly. In some paragraphs, references should be added or revised. Below, I indicated some other moderate to minor issues.*

**AR:** *Thank you for your careful assessment and your constructive feedback. We have investigated the application of the cosmic ray sensor for measurements at a challenging site such as a glacier. The cosmic ray sensor (CRS) that lies below the snowpack has been presented in previous studies (e.g. Kodama, 1980, Paquet and Laval, 2005). A CRS which lies on a large hydrogen pool, such as ice, has been presented by (Howat et al., 2018) where they deployed a CRS on the Greenland ice sheet. To complement these studies, we assessed the application of this device on an Alpine glacier. The primary focus lies on the measurement setup and what new knowledge we gain from such observations. To state this more clearly, we rewrote the end of the introduction and state the study objective more clearly (see below).*

In this study, we investigate the applicability of a CRS installed below the snowpack to derive continuous SWE observations on an Alpine glacier in Switzerland (Glacier de la Plaine Morte). More specifically, we (i) analyse the CRS performance by comparing its SWE estimates to manual field observations. With the continuous observations of SWE and SD we (ii) analyze the evolution of snow density over the course of a winter season including the influence of meteorological conditions. Finally, we use the continuous observations to (iii) assess the performance of scaling readily available precipitation observations of nearby AWS and gridded precipitation data with a constant factor.

**Specific comment**

RC: *Please use the same units throughout the paper. SWE is usually given in mm (or kg/m2), not in cm w.e.*

AR: *We changed the units for snow water equivalent (SWE) to mm w.e., but kept the unit for snow depth (SD) as cm following the unit guidelines in Fierz et al. (2009). We adapted all figures accordingly.*

**Abstract**

RC: *p. 1, l 12-16: The aspect of the comparison of the cosmic ray SWE values and the scaled precipitation is represented quite dominant in the abstract. I think this aspect can be reduced to two 2 sentences and the abstract should better include also a statement on the general applicability.*

AR: *We agree and reduced it to two sentences.*

Moreover, we compare daily SWE amounts to precipitation sums from three nearby weather stations located at lower elevations, and to a gridded precipitation dataset. We determine the best-possible scaling factor for these precipitation estimates in order to reproduce the measured accumulation on the glacier. Using only one scaling factor for the whole time series, we find a mean absolute error of less than 8 cm w.e. for the reproduced snow accumulation. By applying temperature-specific scaling factors, this mean absolute error can be reduced to less than 6 cm w.e. for all stations. The continuous SWE measurements were also used to define a scaling factor for precipitation amounts from nearby meteorological stations. With this analysis, we show that a best-possible constant scaling factor results in cumulative precipitation amounts that differ by a mean absolute error of less than 80 mm w.e. from snow accumulation at this site.

**1. Introduction**

RC: *In general, a statement on remote sensing approaches to derive snow cover properties in alpine areas is missing (e.g. l. 31ff) – please give a short overview on such techniques.*

AR: *We included the following paragraph on remote sensing approaches.*

> Spaceborne sensors can provide observations of snow cover, SWE and SD with a large spatial coverage. However, these observations often have a low spatial resolution and estimates of SWE are affected by snow properties such as the snow crystals and the liquid water content (Clifford, 2010, Dietz et al., 2012). In addition, uncertainties are increased for complex topographies (Smith and Bookhagen, 2016) and deep snowpacks (Smith and Bookhagen, 2018).

RC:   *p.2, l.2: Not only the cold and windy conditions are a big challenge for in situ snow measurements in high mountains; please add that they are also often limited by difficult accessibility, complex terrain etc.*

AR:   *We added a further sentence to explain the limited accessibility and complex terrain.*

> In particular, the cold and windy conditions pose the main challenge for accurate measurements (Sevruk et al., 2009, Rasmussen et al., 2012, Kinar and Pomeroy, 2015). The complex topography and limited accessibility add further challenges in high mountain regions.

RC:   *p.2, l.13-21: The statements in this paragraph should be revised carefully as some statements are not correct. Some explanations are given in the following: Schmid et al. (2014) combined the upGPR with a snow depth sensor to additionally derive the liquid water content in snow. The main reasons for combining upGPR travel time with GPS signal strength in Schmid et al. (2015) were to eliminate an overestimation in snow depth during wet snow conditions, which would be the case by using only upGPR measurements, and to be independent of poles as both sensors were buried beneath the snowpack, which could be useful, e.g., in avalanche prone slopes. Moreover, with this upGPR-GPS sensor combination it was possible for the first time to derive SWE, snow height and liquid water content simultaneously. Schmid et al. (2015) is not suitable as reference in l.20 and Heilig et al. (2009) not for SWE measurements. Besides citing Steiner et al. (2018), Henkel et al. (2018, TGRS) and Koch et al. (2019, WRR) should be added as references in l.20. Besides snow accumulation, the GPS techniques derive snow properties under snow ablation/melt conditions. Additionally, in the latter reference, it was possible to derive three snow cover properties (SWE, snow depth and LWC) simultaneously with only one sensor setup.*

AR:   *We revised this paragraph carefully, and changed it as follows.*

> ~~Ground-penetrating radar (GPR) is another method to determine snow accumulation and has been used in various studies (e.g. Heilig et al., 2009, 2010). Schmid et al. (2014) combine a snow depth (SD) sensor with an upward looking GPR (upGPR) installed within the ground below the snowpack. This combination results in continuous estimates of liquid water content, SD and SWE at a high temporal resolution. SWE derived from this method lies within ±5% discrepancy from manual measurements. In a follow-up study, Schmid et al. (2015) combined an operational upGPR with a low-cost GPS to render the approach independent from additional sensors (e.g. SD). Despite the good agreement with manual measurements of SWE, the underlying algorithm to derive SWE from the upGPR is still prone to errors. For instance, a deviation of 10% in SD may lead to an over- or underestimation of 30-40% of the resulting SWE (Schmid et al., 2014). Furthermore, erroneous identifications of the reflection horizons affect the resulting SWE (Heilig et al., 2009; Schmid et al., 2015).~~
>
> Other in situ devices include ground-penetrating radar (GPR) and sub-snow GPSs. Upward-looking GPR systems are installed below the snowpack and provide information about the snow stratigraphy (Heilig et al., 2009) and snow depth (SD, Heilig et al., 2010, Schmid et al., 2014). Combined with a low-cost GPS, Schmid et al. (2015) derived the liquid water content, SD and SWE independently from additional information and mast poles, making the system suitable for avalanche-prone slopes. Recent studies present sub-snow low-cost GPS as a promising method to continuously derive SWE (Steiner et al., 2018, Henkel et al., 2018, Steiner et al., 2019, Koch et al., 2019). This method uses two GPS antennas one of which is placed below and the other above the snowpack. Because the GPS signals are influenced when traveling through the snowpack, the difference in received signals can be used to quantify SWE, SD and liquid water content. GPS signals are freely available but the signal strength may be limited in high mountain regions depending on slope exposition and location (Koch et al., 2019).

**RC:** *p.3, l.23: I would not name it in a second application. This is rather a further type of validation (besides your manual SWE measurements) for CRS SWE.*

 **AR:** *In general, we have more confidence in the SWE observations by the CRS than in the scaled precipitation measurements. For this reason, we use SWE to find the optimal scaling factor for precipitation and thereby assess an approach that has been previously applied.*

**2. Study Site & 3. Data**

**RC:** *I would suggest to merge sections 2 and 3.*

 **AR:** *We added the section "Study site" under the Section "Data". In the revised manuscript, Section 2 is named "Study site and data".*

**RC:** *p.4, l.2: Although you have mentioned the altitude of your study site in the introduction, this should definitely also be mentioned in this section.*

 **AR:** *We agree and modified the paragraph as follows.*

> Our study site is located on the Glacier de la Plaine Morte (in the following: Plaine Morte) in Switzerland, where we deployed a subsurface CRS along with an  AWS at an elevation of 2690 m a.s.l. (Fig. 1). Plaine Morte is situated on the ridge between  the Bernese Alps in the North and the Rhône valley in the South (Huss et al., 2013) and is surrounded by mountain peaks with elevations from 2926 m.a.s.l (Pointe de la Plaine Morte) up to 3244 m a.s.l. (Wildstrubel, see Fig.1).

RC: *p.4, l.10f: How fast does the glacier move? Is there an effect on the measurements (e.g. on the SD sensor installed on a pole)?*

AR: *We added the following part to the section "Study site" to answer the question about the surface velocity. Concerning the second question, we did not add an explicit statement on this because we expect an effect of the glacier movement to be small. The AWS was designed not to be affected by glacier movement or ice melt. As shown in Fig.1d, it has three large wooden beams as a foundation of the main pole. We do address the influences of this mast design on the SD measurements in Section 5.2.*

> [...] the winter snow distribution shows only a small spatial variability (GLAMOS, 2018) and the surface velocity is low (2-5 m per year according to Huss et al., 2013).

**4. Methods**

RC: *I would suggest including Subsection 4.1 in Section 3.*

AR: *We understand this point, but we decided against including subsection 4.1 into Section 3 because we consider it a method rather than data.*

RC: *The title of Subsection 4.2 might be misleading – it would be better to directly refer to CRS SWE and generally separate between SWE and snow density derivation.*

AR: *We split this section into two parts as suggested. In the revised manuscript the section names are as follows: 3.2 Calculating SWE from neutron counts, 3.3 Calculating snow density and daily changes in SWE, SD and snow density.*

RC: *p.7, l.26: Please insert a reference for the empirical parameters.*

AR: *The empirical function and its parameters have been provided by the manufacturer and previously used by Howat et al. (2018). We added this reference accordingly.*

RC: *p.8, Table 3: Not sure if it really makes sense and is sound to use for the gap filling different meteorological parameters from different stations (e.g. temperature from station a, humidity from station b etc.). In my opinion, rather one station with an overall best fit of all parameters should be used. Please state on this.*

AR: *We understand this point of view. With regard to physical consistency and in, for example, an modeling application this aspect is crucial. In our study, however, we have independent SWE measurements which were not affected by the measurement gap of the station. To complete the bulk snow density evolution, we considered snow depth as the most important parameter and chose the best correlation for it (IMIS station SLFGA2). The correlation of temperature and relative humidity are similar for both, the chosen IMIS station (SLFDIA) and SLFGA2. For wind speeds, in contrast, the correlation is significantly smaller for SLFGA2 or SLFDIA compared to the chosen station (SLFGU2). In general, our results of the prevailing meteorological conditions during process-dominated days would remain similar. Figure 8a and b would remain similar because of the similar correlation. Figure 8d would also not be affected because wind is not displayed during the gap-period as we did not fill the data gap of wind direction.*

RC: ***p.8, l.1: Please use just one unit for SWE (either mm or kg/m2). Regarding SD – in the figures, you use [cm] and here you define SD in [m] – this should be uniform throughout the paper.***

AR: *We adapted the units to mm w.e. for SWE and cm for SD following the guidelines suggested by Fierz et al. (2009). To make the equation consistent with the given units, we added a conversion constant (c) with a value $100\,cm\,m^{-1}$ to the equation.*

> The bulk snow density ($\rho_{\overline{crs,sr}}\rho_{crs\_sr}$, in $\mathrm{kg\,m^{-3}}$) is  derived from daily SWE  ($SWE_{crs}$, in mm w.e. or $\mathrm{kg\,m^{-2}}$, Fierz et al., 2009) and daily SD measurements ($SD_{sr}$, in cm) according to
>
> $$\rho_{\underline{crs,sr}crs\_sr} = \frac{SWE_{crs}}{SD_{sr}} \cdot c \tag{1}$$
>
> with c equal to $100\,\mathrm{cm\,m^{-1}}$ to assure unit consistency.

RC: ***p.9, Fig.2.a: Actually, no red or black crosses are visible in the figure (only red and grey horizontal lines) – please state on this and/or correct. Moreover, the error bars are not really readable. A revised version of this figure would be helpful (it could make sense to display the error bars in a separate figure).***

AR: *The crosses were not visible because the scale is too large for the uncertainties to be visible. We replaced Figure 2 with a new figure (see below). The uncertainties of SWE measurements are discussed and presented in a new section (Section 3.4 Estimating the uncertainty of CRS).*

[Figure]

Figure 2: Relation between SWE and the neutron count rate. Grey dots represent the uncorrected hourly neutron counts and black dots the uncorrected daily means. The orange dots represent the corrected daily means. Red dots show SWE from the field data and the corresponding neutron counts of the field work days.

**RC:** *p.10, l.8: Please introduce N or do you mean Ni?*

 AR: *N refers to $N_i$. We corrected it throughout the manuscript.*

**RC:** *p.10, l.10: Why did you chose +/- 1cm? Can this be underlain with a reference?*

 AR: *The systematic bias of +/- 1 cm originates from an analysis during snow free conditions (not shown), so it cannot be underlain with a reference. But we changed the uncertainty estimate of the SWE observations as suggested by referee #2 and documented it in Section 3.4 Estimating the uncertainty of the CRS. The new approach is based on error propagation of a non-linear equation and contains no additional systematic bias anymore.*

**RC:** *p. 10, l.19: In an earlier section you mentioned 4.8 m instead of 4.75 m – please unify.*

 AR: *We changed it to 4.8 m consistently.*

**5. Results**

**RC:** *p.14, Fig.3: Please describe the vertical dashed lines in the figure caption or in a legend. In general, this figure would benefit to be displayed larger (if possible).*

**AR:** *We replaced this figure with an improved figure and adapted the figure caption (see below). In the revised manuscript, Figure 3 has become Figure 4.*

[Figure]

Figure 4: Continuous observations of (a) SWE, (b) SD and (c) snow density with their daily standard deviation. The red dots show the manual field measurements with their uncertainties (salmon bars). The dotted (dashed) line shows the day of the seasonal maxima in SD (SWE).

**RC:** *p.15, Fig.4: I really like this figure!*

**AR:** *Thank you.*

**RC:** *p.17, l.22: You should underlie the statement of rain gauge undercatch with a reference.*

**AR:** *In the revised manuscript, we added references for when we refer to undercatch by rain gauges. In the following, two relevant manuscript excerpts are shown.*

> [Section 4.3.] Without applying a scaling factor, we see a large difference between cumulative precipitation and snow accumulation on the glacier (Fig. 9). This could be due to the high spatial variability of solid precipitation and/or undercatch of rain gauges (Kochendorfer et al., 2017, Pollock et al., 2018).
>
> [Section 5.3] A drawback for AWS stations is the potentially large undercatch of solid precipitation combined with high wind speeds which can be on the order of a factor of three given solid precipitation and high wind speeds (Kochendorfer et al., 2017).

**6. Discussion**

**RC:** *Please add the following points in the discussion: Is there a general SWE limit by using CRS? How big is the footprint of the sensor and which shape does it have (e.g. conical)?*

**AR:** *For the general SWE limit no distinct value can be given because the relation between neutron counts and SWE is of an exponential nature. We added the following paragraph in Section 5.1 to address this point. To assess the footprint of a CRS lying below the snowpack is beyond the scope of this study. The dispersion and production of fast neutrons within the snowpack remains unclear and would require an in-depth investigation with a different study setup. Moreover, it probably would also require the modeling of neutron trajectories. In the new manuscript, we suggest an investigation on the footprint as a potential future study.*

> In the second winter season, SWE amounts were exceptionally high with more than 2000 mm w.e. Nevertheless, the agreement to field measurement is within ±10% indicating that the measurement limit of SWE has not yet been reached. Due to the exponential nature of the relationship there is no distinct threshold beyond which the relative neutron count is no longer sensitive to SWE (Fig.2).

**RC:** *p.22, l.12: Please specify why there might be problems between 90 and 120 cm.*

**AR:** *We have removed this part from the manuscript because an explanation is too speculative. Moreover, we only apply the equation provided by the manufacturer. An investigation of further relations between neutron counts and SWE might help explain such discrepancies at these particular SWE amounts but is beyond the scope of this study. In general, our results show that the manual measurements are in good agreement with the given conversion equation (neutron counts - SWE, Fig.2). But, we discuss the potential influences of how we process raw neutron counts in the revised manuscript.*

**RC:** *p.22, l.26-30: Please insert references in this paragraph.*

**AR:** *We re-wrote the discussion on the CRS performance and limitations and rephrased this paragraph with reference to the introduction where all references are included.*

> The main advantage of the CRS is that it can be deployed in an exceptionally wide variety of terrain. There is no need for a stable and flat surface nor does it depend on the reception of satellite signal for its measurements (cf. Section 1.1).

**7. Conclusion**

**RC:** *As this study investigates to a quite big extent the development of the snow density at your study site, this should also be mentioned more prominently in the conclusions section.*

AR: *We added the following paragraph in the conclusions.*

> With the daily mean snow density observations, we showed that the evolution of the bulk snow density can be divided into three main periods; accumulation, densification and ablation. Throughout the accumulation period, snow densities are low with periodical repetitions of snowfall and subsequent densification. At the seasonal maximum of SWE the snowpack densifies during several days before its melting period begins. Additionally, we investigated these three processes at a daily basis and could attribute general meteorological conditions to each process.

**Appendix A**

**RC:** *In my opinion the appendix should be integrated in the methods section.*

AR: *We agree and integrated it in Section 3.2 Calculating SWE from neutron counts.*

**RC:** *p.25, l.12: Please introduce N also in the text.*

AR: *We dedicated a own section (3.2) on how we process neutron counts and adapted the variables accordingly.*

**Author Response to Reviews of**

**Evaluating continuous and autonomous snow water equivalent measurements by a cosmic ray sensor on a Swiss glacier**

Rebecca Gugerli, Nadine Salzmann, Matthias Huss and Darin Desilets

*The Cryosphere Discussion,* `doi:10.5194/tc-2019-106`
* * *
**RC:** *Reviewer Comment*,     AR: *Author Response*,     ☐ Manuscript text

**Anonymous Referee #2**

We would like to thank the anonymous referee #2 for his/her time and the thoughtful and constructive review, which significantly improves our manuscript.

Following the suggestions by referee #1 and #2, we decided to change the title of the paper to "Continuous and autonomous snow water equivalent measurements by a cosmic ray sensor on an Alpine glacier".

**General comments**

**RC:** *This paper evaluates the snow accumulation on the Plaine Morte glacier by means of a buried cosmic-ray neutron probe (CRNS) and an approach based on the scaling of the precipitation records of nearby meteorological stations. The accuracy of the field data is assessed by the propagation of possible error sources. Together with the combined approach using different types of field data, this gives important insights into the evolution of the snow pack on the glacier. The language of the paper is appropriate, as are the figures and tables. Partly, the paper would benefit from considering a geographically broader view on the state-of-the-art as many references focus on Switzerland. In principle, the paper is suitable for publication in this journal. In particular, the added value of the paper lies in applying a buried CRNS together with other measurements for continuously monitoring the snow accumulation of a mountain glacier.*

**AR:** *Thank you for your valuable assessment and the interesting feedback. We admit that the state-of-the-art has many, but not only Swiss references. In the revised manuscript, we broadened the introduction and included more more non-Swiss references.*

**RC:** *However, prior to further consideration for publication, the following two major concerns need to be addressed carefully:*

**AR:** *To address these two major concerns in more detail, we split the following comment of referee #2 into smaller parts. That allows us to directly address each raised point.*

**RC:** *(1) The story line of the paper needs to be clarified. The title and the final conclusions do not match well with the analysis made. Furthermore, the second part of the analysis is not (yet) connected to the rest of the paper. One could think of some logical links between the two parts, but it is important to state this more clearly, and to frame the rest of the paper accordingly. In addition, it would help the reader if the*

*novelty would be more pronounced in the abstract and the conclusions.*

AR: *The main focus of the paper is to assess the application of a cosmic ray sensor (CRS) in combination with the sonic ranging sensor for continuous snow water equivalent (SWE) and snow depth (SD) measurements on Alpine glaciers, and to show the advantages of such a measurement setup. We revised the story line of the paper and state our study objectives and the link between precipitation scaling and the CRS measurements more clearly.*

RC: **(2) While the error propagation of the snow depth, snow density and the meteorological measurements is reasonable and covers all important sources of uncertainty, this is not the case for the CRNS data. Most notably, the instrument's precision is most likely largely overestimated. Furthermore, a decrease of the error with increasing SWE is highly unlikely with mostly likely the opposite behavior being the case. Currently, only the uncertainty of the neutron count rate is considered, and a constant error is added despite the high non-linearity of the signal. The latter is probably the reason why the relative accuracy seems to increase with higher snow accumulation values. The statistical error of neutron count rate itself is an important element of measurement uncertainty, but it refers to uncorrected variations only. The uncorrected count rate includes variations not only of the accumulated SWE but also variations of incoming neutrons, atmospheric pressure, and in atmospheric moisture.**

AR: *We addressed this major concern in two ways. First, we approximate the precision by means of error propagation of a non-linear equation. Thereby, we take all corrections of the raw neutron count rate into consideration. With this approach, we also determine the driving uncertainty for the precision and present it in Figure 3b. The absolute precision decreases with increasing SWE. We calculated the precision for two temporal resolutions and show that the precision is considerably lower at the hourly resolution compared to the daily resolution. The calculation of the precision is documented in Section 3.4 of the revised manuscript and presented in the following.*
* * *
**3.4 Estimating the uncertainty of the CRS**

The calculated SWE is determined by the corrected neutron count relative to when the CRS is uncovered by snow ($N_{\text{rel},i}$, Eq. 4). We base our error propagation on all corrections applied to the raw neutron count. Therefore, we assemble Eq.1-4 into

$$N_{\text{rel},i} = N_{\text{raw},i} \cdot (\beta \cdot (\frac{F_{\text{inc},i}}{F_{\text{inc},0}} - 1) + 1) \cdot exp\left(\frac{p_i - p_0}{L}\right) \cdot \frac{1}{N_0} \tag{8}$$

The raw neutron count ($N_{\text{raw},i}$), the incoming neutron flux ($F_{\text{inc},i}$) and air pressure ($p_i$) change with time, but remain independent from each other. Following the rules of error propagation of a non-linear equation, we approximate the uncertainty in $N_{\text{rel},i}$ as

$$\sigma^2_{N_{\text{rel},i}} \approx \left(\frac{\partial N_{\text{rel},i}}{\partial N_{\text{raw},i}}\right)^2 \cdot \sigma^2_{N_{\text{raw},i}} + \left(\frac{\partial N_{\text{rel},i}}{\partial N_0}\right)^2 \cdot \sigma^2_{N_0}$$

$$+ \left(\frac{\partial N_{\text{rel},i}}{\partial F_{\text{inc},i}}\right)^2 \cdot \sigma^2_{F_{\text{inc},i}} + \left(\frac{\partial N_{\text{rel},i}}{\partial F_{\text{inc},0}}\right)^2 \cdot \sigma^2_{F_{\text{inc},0}} + \left(\frac{\partial N_{\text{rel},i}}{\partial \beta}\right)^2 \cdot \sigma^2_{\beta}$$

$$+ \left(\frac{\partial N_{\text{rel},i}}{\partial p_i}\right)^2 \cdot \sigma^2_{p_i} + \left(\frac{\partial N_{\text{rel},i}}{\partial p_0}\right)^2 \cdot \sigma^2_{p_0} + \left(\frac{\partial N_{\text{rel},i}}{\partial L}\right)^2 \cdot \sigma^2_L \tag{9}$$

The uncertainty $\sigma^2_{N_{\text{rel},i}}$ is then propagated through Eq.5 to estimate the uncertainty $\sigma_{\text{crs},i}$

$$\sigma_{\text{crs},i} \approx \sqrt{\left(\frac{\partial SWE_i}{\partial N_{rel,i}}\right)^2 \cdot \sigma^2_{N_{rel,i}}} \tag{10}$$

Since the uncertainties are not always known, we assume rather generous estimates for the uncertainties of all correction factors. Table 5 provides an overview of uncertainty estimates for all components.

For all neutron count rates ($N_{\text{raw},i}$, $N_0$, $F_{\text{inc},0}$, $F_{\text{inc},i}$), we assume poissonian counting statistics, which gives the uncertainty as the square root of the neutron counts (e.g. Zreda et al., 2012). With the integration over a time period $t$, the uncertainty is reduced by $t^{-0.5}$ (Schrön et al., 2018). While the relative uncertainty in $N_{\text{raw},i}$ varies between 1.5%-5.3% for hourly observations, it varies between 0.3%-1% for the integrated daily estimates of our study.

The incoming radiation measured at Jungfraujoch has a low statistical uncertainty as its precision is high with around 190 counts per second. However, incoming radiation is corrected by an adjustment factor ($\beta$, Eq. 2) which is rather small for our site. Therefore, we assume also a small uncertainty of 0.03 for $\sigma_\beta$.

The uncertainty in air pressure ($\sigma_{p_i}$, $\sigma_{p_0}$) is based on the instrumental precision of 0.1 hPa (Lufft, 2019) . For the mass attenuation length $L$, we use 132 hPa. An applied uncertainty of of $\pm 2$ hPa corresponds to the difference of shielding depths from latitudes north and south of Switzerland as shown in Fig.1 of Andreasen et al. (2017).

To render the error propagation more robust, we calculated $\sigma_{\text{crs},i}$ using two different time resolutions. We additionally created a synthetic data set for both time resolutions. For the synthetic data set, we varied the time-dependent variables ($N_{\text{raw},i}$, $p_i$, $F_{\text{inc},i}$) uniformly within their observed minima and maxima values. At the hourly resolution it encompasses $4.8 \cdot 10^5$ hours and at the daily resolution it encompasses $4.8 \cdot 10^5$ days.

Figure 3a and b show the resulting precision for an hourly and daily resolution, respectively. Figure 3c and d show the relative contribution of every uncertainty term in Eq. 9, i.e. a high relative contribution indicates that the given parameter is an important source for the overall uncertainty of SWE. Figure 3 shows that the main uncertainty can be attributed to the neutron count uncertainty, independently of the time resolution. However, the precision estimate presented here does not include the uncertainty of the correction parameterization (Eq. 2 and Eq. 3) or the conversion equation (Eq. 5) and its parameters (Table. 4).

Table 5: Compilation of all direct observations and constants as well as the associated uncertainties $\sigma$ at the hourly and daily scale. The units cph and cps stand for counts per hour and second, respectively. Brackets show the minimum and maximum within the time series.

| Variables | hourly values | $\sigma$ (hourly) | $\sigma$ (daily) |
|---|---|---|---|
| $N_{\mathrm{raw},i}$ | $[354; 4450]$ cph | $\sqrt{N_{\mathrm{raw},i}}$ cph | $\sqrt{\frac{N_{\mathrm{raw},i}}{24}}$ cph |
| $N_0$ | 4143 cph | 64 cph | 13 cph |
| $F_{\mathrm{inc},i}$ | $[184; 195]$ cps | $\sqrt{\frac{F_{\mathrm{inc},i}}{3600}}$ cps | $\sqrt{\frac{F_{\mathrm{inc},i}}{86400}}$ cps |
| $F_{\mathrm{inc},0}$ | 191 cps | 0.2 cps | 0.1 cps |
| $\beta$ | 0.95 | 0.03 | 0.03 |
| $p_i$ | $[708; 747]$ hPa | 0.1 hPa | 0.1 hPa |
| $p_0$ | 739 hPa | 0.1 hPa | 0.1 hPa |
| $L$ | 132 hPa | 2 hPa | 2 hPa |

[Figure]

Figure 3: Precision of SWE calculated by means of error propagation. (a) and (b) show the absolute precision with grey dots as an synthetic data set and black dots as the in situ observations. (c) and (d) show the relative contribution of each parameter to the overall precision. (a) and (c) present the results based on hourly observations while (b) and (d) show the results of the daily observations.

**RC:** *An error propagation should thus include the uncertainty of (1) the neutron count uncertainty as already done, (2) the uncertainty of the measurements used for the corrections (Jungfraujoch neutron monitor data, atmospheric pressure, atmospheric moisture), (3) the uncertainty in the parameterisation of the correction functions (e.g., the value for the attenuation length, which may vary in space and time), and (4) the uncertainty in the (not well documented) empirical function relating neutron counts to SWE. In total, from figure 2 the error seems to be rather in the range of 10 to 20% (and thus around ten times larger than estimated in the paper!), with an increasing trend for high SWE values. Also the comparison with the manual measurements (figure 3) shows that the SWE from CRNS is mostly only touching the uncertainty bands of the manual measurements, while is partly entirely off.*

AR: *In the revised manuscript, we present an error propagation considering not only the neutron count uncertainty but also the uncertainty of the measurements used for the corrections. The uncertainty from the parameterisation of the correction function and the empirical function are not included in the calculations of Section 3.4 but clearly stated in this section and in the discussion. The relevant excerpt of the discussion section is shown in the following.*

**5.1 CRS performance and limitations**

The data processing of the neutron counts as presented is straightforward. Given the transformation equation, only the initial neutron count rate can be calibrated. But a variation of this calibration parameter within its uncertainties has little influence on the resulting SWE amounts, especially for amounts larger 400 mm w.e. This is a consequence of the exponential nature of the conversion equation (Eq. 5). More importantly, the neutron count rate may also be influenced by how we correct for air pressure and solar activity even though we apply the same equations as presented in previous studies for SWE (e.g. Howat et al., 2018) or soil moisture studies (e.g. Zreda et al., 2012, Andreasen et al., 2017). In contrast to previous studies of above-ground CRS, we do not correct for the changes in atmospheric moisture. We assume that for the below-ground CRS, fast neutrons are produced within the snowpack rather than in the atmosphere, an assumption also made in Howat et al. (2018), and implicitly made by preceding authors in their studies (e.g. Kodama et al., 1979, Paquet and Laval, 2005, Gottardi et al., 2013). Another source of uncertainty is the semi-empirical fit that has been used in this study. Because our study focuses on the application for snow and glacier studies, we have chosen to apply the relations used by Howat et al. (2018). In general, the conversion function has the potential to introduce considerable uncertainty in the inferred SWE. However, the applied empirical relation has shown to be adequate as the resulting SWE agrees well with independent field measurements, indicating only a minor bias and a standard deviation for individual observations that lie in the range of the uncertainty of the in situ SWE surveys.

For all correction factors such as air pressure and solar activity, we propagated an estimated uncertainty through all equations and show that the precision is mainly defined over the neutron count rate. Assuming that the parameterization of the correction equations carry no uncertainties, the influences of all other measurements and constant parameters are small. Moreover, an independent study by Howat et al. (2018) quantified a precision of 0.7% of a CRS lying below the snowpack on the ice sheet. Their results, however, are affected by lower in situ air pressure and consequently higher neutron count rate. In addition, Howat et al. (2018) observed lower SWE amounts which places them on a steeper part of the calibration curve (Fig. 2). For lower SWE amounts, changes in neutron counts are more sensitive and have a higher precision. The precision can be increased by integrating over longer time periods.

RC: *With the current focus of the paper the lack of a proper error propagation of the CRNS data constitutes a severe issue, as the evaluation and the precision of the CRNS are stated prominently in the title and conclusions. Still, it is interesting to see the application of CRNS for glacier monitoring and I agree with the authors that it constitutes a very promising technique for continuous accumulation measurements on glaciers. Existing uncertainties should, however, be kept in mind instead of propagating an unrealistically high precision of the SWE estimate. I believe there are two equally legitimate strategies on how the authors could address this. One is a true and rigorous error propagation with regard to all relevant uncertainty sources of the CRNS SWE estimate. Another could lie in drawing the reader's attention to the fact, that the uncertainty range could be substantially (up to ten times) larger, combined with reframing the paper towards the application rather than the error propagation.*

AR: *We addressed the major concern of refree#2 in two different ways. First, we calculated a precision based on the neutron count rate and the correction measurements. Second, we draw the readers attention to all uncertainties related to the processing of the raw neutron count in the discussion. Since the main focus of the paper is the application of a CRS, an in-depth investigation of the uncertainties in the correction parameterizations and the semi-empirical conversion equation would be beyond the scope of this study. We adjusted the title and story line of the paper accordingly.*

**Specific comment**

RC:   *Page 3/ Line 33-34: Check the sentence ("..define three different scaling factors, one for..."?).*

AR:   *This part was rewritten and does not include the specific sentence anymore.*

RC:   *Page 4/ Line 2: It would be helpful when the elevation of the glacier and the surrounding mountain peaks would be added here.*

AR:   *We modified the beginning of Section 2.1 Study site as follows.*

> Our study site is located on the Glacier de la Plaine Morte (in the following: Plaine Morte) in Switzerland, where we deployed a subsurface CRS along with an  AWS at an elevation of 2690 m a.s.l. (Fig. 1). Plaine Morte is situated on the ridge between  the Bernese Alps in the North and the Rhône valley in the South (Huss et al., 2013) and is surrounded by mountain peaks with elevations from 2926 m.a.s.l (Pointe de la Plaine Morte) up to 3244 m a.s.l. (Wildstrubel, see Fig. 1). With a surface area of 7.4 km$^2$  and a particularly low elevation gradient, Plaine Morte is the largest plateau glacier in the European Alps. Most of its surface is located between 2650 m a.s.l. and 2800 m a.s.l. (GLAMOS, 2018).

RC:   *Page 5/ Line 18: Can you add a few key facts on how the gridded products is produced. Does it contain station data? If so, how reliable is it when the nearby stations have data gaps?*

AR:   *We added the following paragraph to Section 2.2.*

> The gridded precipitation product, RhiresD, uses rain-gauge measurements from around 400 automatic as well as manual observations. These observations (not available in real time) are quality-checked prior to their processing. The observations are spatially analysed, pre-processed and interpolated to a 1×1 km grid at daily resolution covering the Swiss territory (MeteoSwiss, 2013). The main sources of uncertainty arise from the interpolation, the rain-gauge measurements, the grid spacing and its effective resolution, and the temporal variation of the number of stations. For further information, the reader is referred to the technical document provided by MeteoSwiss (MeteoSwiss, 2013). We extracted daily precipitation estimates of the three grid points closest to the position of the CRS (Table 2 and Fig. 1c).

RC:   *Page 6/ Table 2: Think of readers that are not familiar with the Swiss coordinate system. I would recommend converting the station coordinates into a globally used system like UTM or WGS84 (lat/lon). In any case, add also the EPSG-code of the coordinate system.*

AR:   *We added the WGS84 coordinates of the AWS and the RhiresD in the corresponding table. For RhiresD, we kept the Swiss coordinates to help the reader find the grid cells in Fig.1c. Additionally, we added a cross in the lower left corner of Fig.1c with the corresponding WGS84 coordinates.*

RC:   *Page 7 Line 1: The reliability of the CRNS is one of the objectives, thus could cannot be claimed beforehand.*

AR:   *We changed the paragraph as follows.*

Once deployed, the CRSmeasured reliably The CRS, in contrast, measured continuously over the two winter seasons with one exception . During the exception of a short period end of April 2018, the CRS measured irregularly because of a problem with the connector. However2018. After fixing a faulty connection, the CRS then continued measuring without our interference. In summer 2018, we changed the connector and measurements have been without gaps since. need for further maintenance.

**RC:** *Page 23/ Line 2: the effect is related to SWE not to density.*

AR: *We split the discussion section into two parts and rewrote both which changed also this sentence.*

**RC:** *Page 23/ Line 8: Here, too much confidence is set into CRNS.*

AR: *We rewrote this section and state all influences of snow density estimates more clearly. In the following the corresponding revised paragraph of the manuscript is presented.*

[...] The high snow densities presented here could be a result of changes in the snow physics, measurement errors of SWE and SD estimations (Eq. 7), or a combination of both. Physical changes within the snowpack could be due refreezing of liquid waterat several layers within the snowpack , water saturated snow in the top layers, locally thick ice lenses, or accumulation of liquid water around the CRSwhich eventually refreezes. With the CRS and the SR, we can only determine a mean snow density. Therefore, not all these effects would be identifiable, and explanations remain speculative. Despite all potential explanation for errors by the CRS, it could also be a problem with the SD measurements rather than the SWE measurements. In our study setup, several reasons could cause erroneous SD measurements . FirstlySWE from the CRS could, for example, be affected by a supraficial pond in the vicinity of the site. It remains unclear how such a hydrogen pool would influence the in situ point measurements of the below-ground CRS. Other influences could come from the correction factors of the neutron count rate or the conversion equation applied in this study (cf. Section 5.1). The SD measurements are also susceptible to errors. For example, the snow area below the sonic ranging sensor may show a small depression because of wind turbulence caused by the mast. FurthermoreAdditionally, the snow around the metal mast main pole of the station melts faster possibly leading to a depressionwith a larger radius around the mast. It remains difficult to assess whether the radius of this depression would be within the footprint of the sonic ranging sensor. Nevertheless, these two effects may superimpose. SecondlyIn winter 2017/18, the solar panels were submerged below the snow. To ensure further power supply, we had to free the solar panels by digging a dig them out. This snow pit around the mast in winter 2017/18. This snow pit main pole would have been refilled by wind, but densities are different, probably causing accelerated melt rates around the mast. Thirdly, the influences For more shallow snowpacks, the metal anchorage of the mast's foundations , the wooden beams with the metal anchorage, may cause erroneous SD measurementsfor more shallow snowpacks. This also becomes clear since SD measurementsnever reach might interfere with the SD measurements. The SD measurements, for instance, never observe a SD of 0 cm even though the sensor is calibrated for the mounted height and agreements to snow probings agree during the season (Fig. 4). b and Fig. 6).

*To underline this statement, we show a photo from the mast taken in June 2019. We did not include this photo*

*in the revised manuscript. Between the last field work in April 2019 and this one, the site was not visited, and we encountered it with the large depression around the mast itself.*

[Figure]

Figure 4: Photo of the mast installation taken in June 2019.

[revised manuscript text omitted]
. If we only consider changes in SD, such as the accumulation days defined in Table 6, no increases in SD occur from mid of May to end of May (Fig. 7b) . During this period, most precipitation events are classified as either liquid or. From mid of March to mid of April 2017, some accumulation events result in increases of SD below 10 cm. These events do not have a signal in the SWE measurements~~ precipitation, the time period in which precipitation is accumulated could be adjusted. However, an adjustment of the time period would only partly exclude such events.

The choice of the precipitation data and AWS is also important. RhiresD has shown a better performance especially for the phase-dependent scaling factors. Tsanfleuron (2052 m a.s.l.) has the lowest constant factor (1.8) and MAE ($70\pm37$ mm w.e., Table 7) for the phase-independent approach. Adelboden and Montana which are located north and south of Plaine Morte have higher scaling factors than Tsanfleuron. In addition, they are on either side of the Alpine ridge and dominated by different weather regimes which is also confirmed by analyzing the temporal evolution. In winter 2016/17, many events captured by Adelboden are not represented in Montana (Fig. 9a). Nonetheless, Montana does not perform worse than Adelboden with only one constant factor. In the case of the phase-dependent scaling, the performance of Adelboden is significantly improved reducing its MAE by almost a factor of two.

~~Another uncertainty is introduced by the temperature thresholds applied in this approach. Previous studies have shown that the snow-rain threshold varies on a global scale between −0.4°C and 2.4 °C in the Northern Hemisphere (e.g. Jennings et al., 2018) . To refine the temperature thresholds, we need a higher temporal resolution of SWE measurements . For the SD observations, we have an hourly resolution. Given that precipitation falls in its liquid form, it would not be as an increase in snow accumulation but as a decrease.~~ Our calculation was possible only because we had reliable and continuous snow accumulation data. Because the spatial variability of snow accumulation on Plaine Morte is rather low the analysis can be made with a point measurement as a reference. But at high mountain sites with more topographic gradients, the location of the in situ measurement becomes more important which is why a glacier-wide mean is typically used. Another caveat of this assessment is the uncertainty of the CRS measurements which has not been taken into consideration. Nonetheless, the resulting MAE lie within $\pm13\%$ of the average agreement between CRS and within the uncertainty of manual measurements.

In summary, it is possible to infer the temporal dynamics of snow accumulation at a high-elevation site by means of scaled precipitation data. However, at least one in situ observation is required for applying this approach. The choice of the precipitation data series and the time period considered is crucial for this methodology.

**6 Conclusions and perspectives**

During two winter seasons, we observed snow accumulation and ablation on a Swiss glacier at a daily resolution. The deployed CRS withstood the harsh environmental conditions at the high mountain site and measured reliably. The validation with manual field measurements indicated a mean accuracy of +2% 13%. In combination with continuous SD measurements, the CRS provided daily mean snow densities that were within a range of ±8% of manual in situ snow density surveys.

With the daily mean snow density observations, we showed that the evolution of the bulk snow density can be divided into three main periods; accumulation, densification and ablation. Throughout the accumulation period, snow densities are low with periodical repetitions of snowfall and subsequent densification. At the seasonal maximum of SWE the snowpack densifies during several days before its melting period begins. Additionally, we investigated these three processes at a daily basis and could attribute general meteorological conditions to each process.

The deployment of the CRS on Plaine Morte provided continuous observations of SWE that could be used to assess the optimal scaling factor for readily available precipitation data. With the optimal scaling factor, we were able to obtain snow accumulation with a MAE of below 80 mm w.e. However, the performance depends on the choice of precipitation data, the choice of AWS, the date of the manual ground measurement and the time period considered. Scaling precipitation with a phase-dependent factor further improves these results.

In summary, we conclude that the CRS is a highly promising device for observing SWE continuously in cryospheric high alpine environments. Despite its limitations through the level of noise and its precision depending on absolute snow amounts, it is suitable for long-term monitoring of SWE in high mountain regions as well as polar regions. In such areas, its resilience in harsh environmental conditions, its rare need for maintenance (once it is properly running) and its flexibility regarding site topography are convincing. For shallower snowpacks, the temporal resolution can be increased to a sub-daily scale. The presented measurement installation is costly. For

long-term monitoring of SWE, such an extensive installation For this study, we chose an elaborate measurement setup which would not be necessary if only SWE measurements are required.

Future studies could analyze the spatial footprint of a CRS lying below the snowpack. Furthermore, spatial distribution of continuous SWE measurements would allow further understanding of the spatial variability in snow accumulation, solid

5 precipitation, precipitation phases and its relation to snow accumulation. In future, the point-scale footprint of the CRS should be better investigated by modelling of neutron trajectories. It would be particularly important to better quantify the influence of hydrogen pools in close vicinity of a subsurface CRS. More investigations into the location-dependent correction of the solar activity would provide further insights into the applied processing of raw neutron counts. The deployment of additional CRS observations in other high-mountain regions of the Alps would not only give further indications on the suitability of

10 precipitation scaling but also the spatial variability of snow accumulation.

*Data availability.* All observations at the Glacier de la Plaine Morte are available upon request from the first author. In future, it will also be available in an online repository.

**7 Correcting raw neutron counts**

The correction of the raw neutron counts ($N_{raw}$) to account for in-situ influences has been presented in previous studies

15 (Zreda et al., 2012; Sigouin and Si, 2016; Andreasen et al., 2017; Howat et al., 2018). In this study, we apply the same equations. The neutron counts of the CRS are corrected with the solar activity ($F_s$) and the in-situ air pressure ($F_p$, Eq. **??**).

$$N = N_{raw} \cdot F_s \cdot F_p$$

The correction factor $F_i$ is determined as

$$F_s = 1 + \beta \cdot (F_{sol} - 1)$$

[revised manuscript text omitted]

---

## Referee Report (RR1)

**Review of tc-2019-106**

**General Remarks**

The authors made a good job revising the original manuscript. I have no major comments. Some minor comments are listed below.

**Specific Remarks**

P3, L51: I guess EdF is using the data to optimise the generation of hydroelectric power. Please add e.g. "plants" after "hydroelectric power".

P7, L4/5: In the second part of this sentence a verb is missing: "The adjustment is negligibly small because our study site geographically close to the neutron monitor at Jungfraujoch."

P25: I'm fine with the reasoning of this section but the language could be improved. In particular, the frequent use of "our" and "we" sounds rather colloquial.

P25, L63: The precision in Howat et al., 2018 doesn't reflect all sources of uncertainty. Thus the value stated there is probably too optimistic. This should be discussed at this point.

P25, L70: What is the rational of this sentence in the context of the manuscript: "The correction factors for air pressure and solar activity can easily be interpolated."?